cognition/behaviour/evolution

evolutionary comparative thanatology, face perception, face pareidolia

**Author for correspondence:**
André Gonçalves
e-mail: a.gnclves@gmail.com

# Staring death in the face: chimpanzees' attention towards conspecific skulls and the implications of a face module guiding their behaviour

André Gonçalves[1], Yuko Hattori[2] and Ikuma Adachi[1]

[1]Language and Intelligence Section, and [2]Center for International Collaboration and Advanced Studies in Primatology, Primate Research Institute, Kyoto University, 484-8506 Aichi, Japan

Chimpanzees exhibit a variety of behaviours surrounding their dead, although much less is known about how they respond towards conspecific skeletons. We tested chimpanzees' visual attention to images of conspecific and non-conspecific stimuli (cat/chimp/dog/rat), shown simultaneously in four corners of a screen in distinct orientations (frontal/diagonal/lateral) of either one of three types (faces/skulls/skull-shaped stones). Additionally, we compared their visual attention towards chimpanzee-only stimuli (faces/skulls/skull-shaped stones). Lastly, we tested their attention towards specific regions of chimpanzee skulls. We theorized that chimpanzee skulls retaining face-like features would be perceived similarly to chimpanzee faces and thus be subjected to similar biases. Overall, supporting our hypotheses, the chimpanzees preferred conspecific-related stimuli. The results showed that chimpanzees attended: (i) significantly longer towards conspecific skulls than other species skulls (particularly in forward-facing and to a lesser extent diagonal orientations); (ii) significantly longer towards conspecific faces than other species faces at forward-facing and diagonal orientations; (iii) longer towards chimpanzee faces compared with chimpanzee skulls and skull-shaped stones, and (iv) attended significantly longer to the teeth, similar to findings for elephants. We suggest that chimpanzee skulls retain relevant, face-like features that arguably activate a domain-specific face module in chimpanzees' brains, guiding their attention.

# 1. Introduction

> What goes on in the chimpanzee's mind when they see such a sight in the forest? We are able to draw some conclusions based on their behaviour when they encounter dead individuals, but deciphering their actual thoughts remains speculative—Christophe Boesch [1, commenting on a picture of a chimpanzee skull]

Chimpanzees and elephants share some curious traits; they are large-brained, long-lived animals with prolonged development, live in complex societies, are capable of mirror self-recognition and display protracted interest towards injured and dead conspecifics [1,2]. Behavioural responses such as physical manipulation of the corpse, vigils and visitations are strikingly similar among these two taxa [3]. Non-human animal interest in skeletons is not reported in the literature apart from records of osteophagia mostly observed in ungulates where the targets typically include horns, hoofs and long-bones, usually explained as a form of nutrient consumption [4]. Elephants appear to be unusual in showing extended attention to dead conspecifics long after they decompose, often interacting with their skeletons for protracted durations. However, there is one study reporting similar behavioural responses in bovids (*Bos javanicus*) to both conspecific and non-conspecific bones, though neither as marked nor as elaborate as seen in elephants [5].

## 1.1. Elephants: observations and empirical research

Elephant post-mortem attentive behaviour has been documented in all three extant species of elephants [6,–8]. Long known to elephant researchers, interest in dead conspecifics is not only limited to carcases but also extends towards conspecific bones and tusks [9]. During a study of elephant carcase decomposition, the skull was transported as far as 100 m from the original site by other elephants [10]. Noticing this natural propensity towards interaction with conspecific bones by wild African elephant, and inspired by Iain Douglas-Hamilton's [11] so-called 'crude experiments' three decades before, Karen McComb and colleagues [12] devised an experiment to empirically measure how their interest contrasted with other objects positioned in the environment. Their experiments consisted of systematically placing three objects in a line, 1 m from each other, order randomized in each trial and placed at 25–30 m from the nearest elephant group. The first condition consisted of an elephant skull, elephant tusk and a piece of wood. Similarly, in the second condition, the skull of an elephant was presented alongside the skulls of a buffalo and a rhinoceros. Finally, the third condition comprised three skulls of elephant matriarchs, one of an individual known to the group while alive. Because olfaction is a substantial sensory domain in elephants, to control for smell, all bone stimuli were completely dried and treated. The results showed that elephants (i) approached and manipulated elephant tusks significantly more than other objects, (ii) similarly showed more interest in conspecific skulls than non-conspecific skulls, and (iii) appeared not to differentiate skulls of previously known individuals from the skulls of strangers. Their main findings are further supported by an informal experiment conducted by Goldenberg and Wittemyer [9] in which they presented elephant, giraffe and Cape buffalo bones to wild elephants, with the most interest shown to elephant bones. Similar results were obtained in captive elephants (Rassmussen in [9]).

## 1.2. Chimpanzees: observations and empirical research

Two classic comparative psychology studies give us some insights into this topic regarding skeletons. First, Ladygina-Kohts [13] researched the comparative development of chimpanzees and humans. She presented her juvenile male chimpanzee (Joni) with an array of stimuli (dead hen/grouse, dead magpie, dead hare, monkey skeleton and human skull). His general response towards these objects was initially of fear and apprehension and then curiosity and excitement, touching them with his index finger and then smelling them. Second, Hebb's seminal experiments on fear, with 30 captive chimpanzees, involved presenting each of them with several 'fear-inducing' stimuli (juvenile chimpanzee skull with a movable jaw, chimpanzee death-mask, an infant chimpanzee corpse, etc.). Their responses were mostly fearful. Hebb's [14] attempt to explain the underlying fear mechanisms initiated by these objects was an important contribution: the chimpanzees were faced with objects exhibiting conflicting perceptual cues leading to incompatibilities at the cognitive level (incomplete physical features and lack/presence of movement).

Several published reports describe chimpanzees' reactions towards their dead, ranging from affiliative to aggressive and from quiet/passive to loud/expressive [15–22]. Indeed, chimpanzee mothers have been observed carrying their dead infants for days, weeks or months [23], a pattern

commonly observed in many females across the primate order [24,25]. While these publications have contributed to our knowledge of *Pan thanatology*, we still know very little about how chimpanzees engage with skeletons in their natural environments. Not many observations have been made of this phenomenon with one notable exception. Among many thanatological interactions in wild chimpanzees and gorillas, Watts [26] recounts two cases where the Ngogo Community chimpanzees interacted with skeletonized conspecifics (retaining some hair and ligaments). In the first case, the chimpanzees stopped, looked vigilant, gave alarm calls and clustered on the ground and in the trees around the chimpanzee skeleton, looked at it for around 5 min and stayed at 3–4 m from it. In the second case, the chimpanzees clustered around the skeleton, peering at it for 2 min with many positioned 0.5 m from it, before departing. Aware of McComb *et al.*'s [12] elephant study, Watts [26] goes on to conjecture that skeletons capture attention because they still bear some 'iconic resemblance' to living chimpanzees and that individuals may employ anatomical knowledge based on skeletons of prey species or perhaps in conjunction with the recognition of chimpanzee teeth or hair (when present in the skeleton). Notably, both deaths resulted from intra- and inter-community killings, respectively, carried out by some of the chimpanzees involved in these post-mortem interactions. Given the diversity of chimpanzees' thanatological responses [27], we should be careful not to interpret these results conclusively as a general pattern but rather integrate them within a wider range of possible chimpanzee responses to conspecific skeletons.

## 1.3. Face perception research

The literature on chimpanzees' responses towards their dead shows that they (and other non-human primates) pay substantial attention to the face, presumably due to its communicative value [2]. Faces are an important category of visual stimuli for both humans and non-human animals as evidenced by several studies conducted on vertebrate species [28]. They can provide information on identity (sex, age, rank), communicate many emotional expressions and provide information on attentional states (gaze) and attraction (mate quality) [29]. Comparative cognitive research on facial perception has hinted at a close relationship between humans and chimpanzees. Like humans, chimpanzees show similar responses to faces, for example, rapid individuation of faces, conspecific face-inversion effects and second-order relational information [30]. This should not be surprising since there was probably a strong evolutionary pressure for attending to faces in primates (and other taxa), since faces carry all sorts of advantageous information (conspecific versus non-conspecific, in-group versus out-group member, threat versus non-threat, etc.). In non-human primates, such face-to-face interactions start from birth with these experiences shaping their brain activity and guiding their knowledge of socially appropriate behaviours continuously throughout life [31,32].

Research into human face perception accounted for distinct levels of configural/holistic processing since faces carry specific information: *first-order relational properties* (i.e. general arrangement of facial features: eyes above the nose, nose above the mouth, allowing us to detect a face) and *second-order relational properties* (i.e. discrete variation in such facial arrangement, allowing us to discriminate between individual faces: spacing and positioning of eyes, mouth, nose) [33]. Many hypotheses have been put forth to explain this general propensity to attend to faces; one such involves an innate processing module followed by a learning module [34]. While there is debate as to which precise aspects are innate versus learned, general support for both has been shown both in the human and comparative literature [30,35].

## 1.4. Research motivations and questions

We began this study with the assumption that chimpanzee skulls are perceived much like chimpanzee faces and, likewise, would be subjected to identical attentional biases. With different sets of stimuli, controlled for size, in hypothesis 1a, we predicted that chimpanzees would show a conspecific bias that would be stronger for faces, followed by skulls and finally stones. Moreover, assuming such interest is guided by some sort of face-like recognition (i.e. configural facial/face-like arrangements and outlines), in hypothesis 1b, we predicted that chimpanzees would exhibit longer looking times towards conspecific frontal-facing and diagonally presented stimuli, but to a lesser extent for laterally presented stimuli. Moreover, in hypothesis 2, we predicted that chimpanzees would show a preference for chimpanzee faces over chimpanzee skulls and the outlines of skulls (filled with stone textures). Finally, because elephants show significant interest towards tusks, perhaps by association (tusks are visibly salient body parts in live conspecifics), likewise in hypothesis 3, we predicted that chimpanzees would direct their attention most prominently towards the teeth as posited by Watts [26] (figure 1).

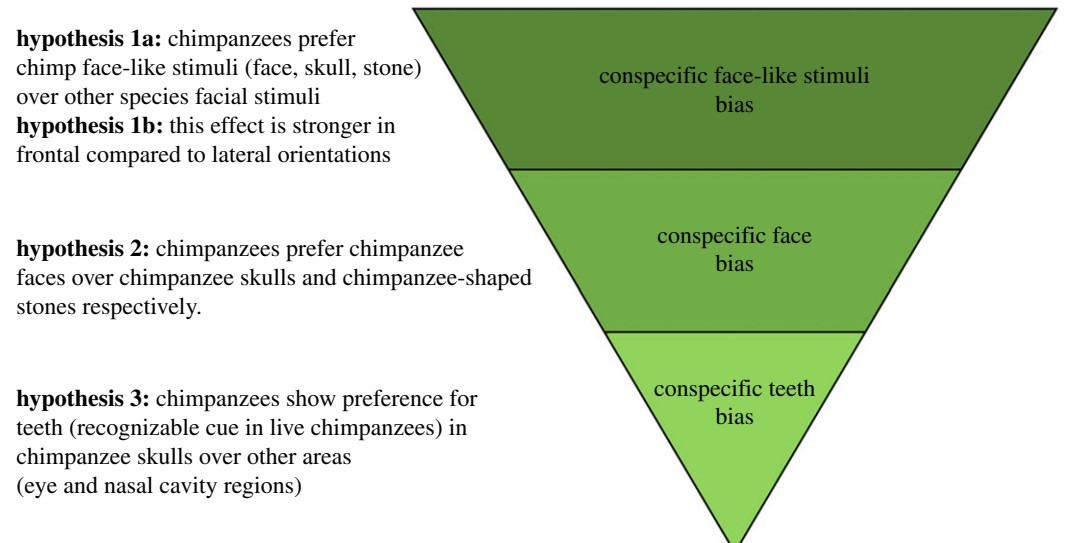

**hypothesis 1a:** chimpanzees prefer chimp face-like stimuli (face, skull, stone) over other species facial stimuli
**hypothesis 1b:** this effect is stronger in frontal compared to lateral orientations

**hypothesis 2:** chimpanzees prefer chimpanzee faces over chimpanzee skulls and chimpanzee-shaped stones respectively.

**hypothesis 3:** chimpanzees show preference for teeth (recognizable cue in live chimpanzees) in chimpanzee skulls over other areas (eye and nasal cavity regions)

conspecific face-like stimuli bias

conspecific face bias

conspecific teeth bias

**Figure 1.** Diagram illustrating our hypotheses from the general (conspecific stimuli), the central (conspecific face) to the narrow (conspecific teeth).

Despite the intriguing findings, McComb *et al.*'s [12] study was not without limitations. For instance, they could not control for parameters such as size, colour or luminance. While the latter two might not be critical to their experiment, given elephants' poorer visual acuity in comparison with chimpanzees' [36,37], the fact that an elephant skull is roughly twice the size of a rhinoceros or a hippopotamus skull remains somewhat more problematic. Because chimpanzees are predominantly visual, to answer these questions, we devised an experiment showing them images, controlled for size and luminance, and measured their differential looking times using an eye-tracking device.

## 2. Material and methods

### 2.1. Subjects

The experiment initially included 10 adult chimpanzees (*Pan troglodytes*), with three subsequent dropouts[1] reducing the total to seven subjects (three males and four females) (table 1; electronic supplementary material, Data). All individuals were housed at the Primate Research Institute, Kyoto University, Japan. These individuals were members of two social groups (totalling 11) living in an environmentally enriched facility comprising two outdoor enclosures (250 and 280 m$^2$), an open-air outdoor enclosure with vegetation and climbing structures (700 m$^2$) and indoor living rooms linked to the testing rooms [39]. They had access to water ad libitum and received a variety of foods several times a day. All research procedures followed institutional guidelines (Primate Research Institute 2010 version of 'The Guidelines for the Care and Use of Laboratory Primates'), and the experimental protocol was approved by the Animal Welfare and Animal Care Committee of the Primate Research Institute and the Animal Research Committee of Kyoto University.

### 2.2. Apparatus

This research was conducted in an experimental booth (1.80 × 2.15 × 1.75 m) inside a testing room. Each chimpanzee voluntarily walked to the booth through an overhead walkway connected to the indoor rooms and outdoor enclosures. We used a Tobii eye tracker (60 Hz; X300; Tobii Technology AB,

---

[1]These dropouts were due to different causes (neurological, physical and temperamental) which affected the completion of the sessions. One chimpanzee (Pendesa) exhibited erratic visual patterns, she suffers from an arachnoid cyst in her brain which seems to have partially affected her visual field (see [38]). Another subject (Mari) has difficulty breathing through her nose which affected her comfort sipping the juice resulting in her ceasing the experiment altogether. The third (Pan), would for no apparent reason (without any overt signs of disgust or distress) abandon the experiment midway through or not come at all despite our best attempts to entice her.

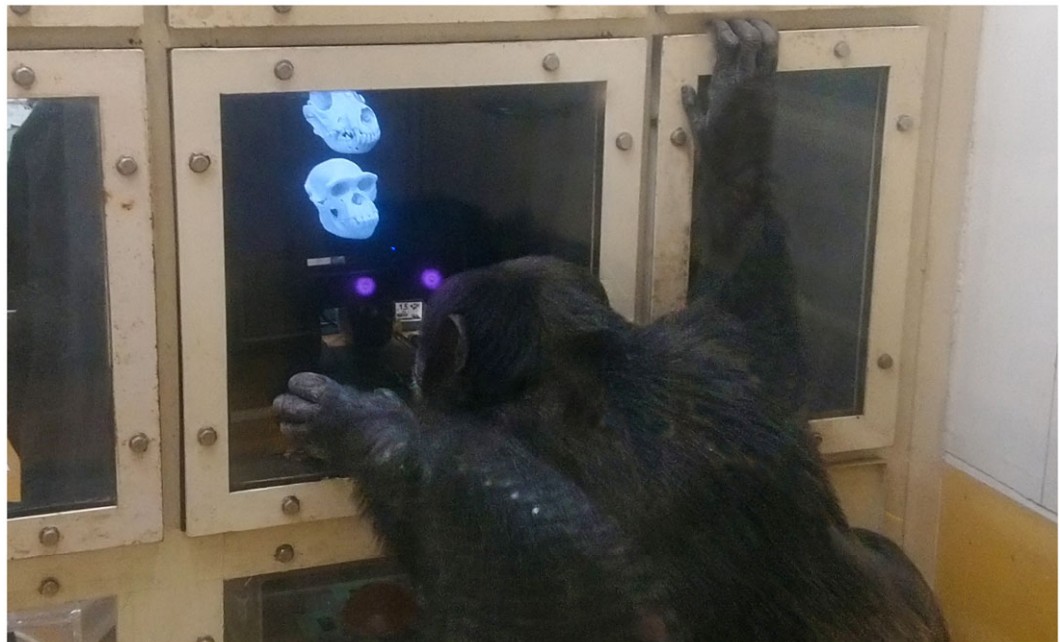

**Figure 2.** One test subject (Ayumu) performing an eye-tracking session in the experimental booth.

Stockholm, Sweden). Sets of images were shown at a resolution of 1280 × 720 pixels on a 23-inch LCD monitor (*ca* 43° × 24°) using TobiiStudio software (v. 3.2.1) at approximately 60 cm. Both the eye-tracking device and the monitor were outside the experimental booth, the subject's eye movements being recorded through a transparent acrylic panel (1 cm thick). To reduce head movements during stimulus presentation, the subjects were able to sip juice through a nozzle and tube attached through a hole in the acrylic panel (figure 2). At the beginning of each test session, automated calibrations were conducted for each subject; these involved one small clip of a stirring object presented twice on each opposing corner of the screen. Following these measures, calibration errors were typically within 1° [40].

## 2.3. Procedure and stimuli

For experiment 1, we presented a total of 180 images to the chimpanzees (4 species × 3 types (material) × 3 angles × 5 image variations). Each image group comprised four species (cat, chimp, dog and rat) shown simultaneously at each corner of the screen (figure 2). Each image group consisted of one of three types (either skull, face or skull-shaped stone). Moreover, each of these conditions was presented in three different orientations (diagonal, frontal and lateral) (figure 2). There were five variations per image within image groups (i.e. five different chimpanzee skulls presented in frontal, diagonal and lateral orientations across five different sessions) (figure 3). No images were repeated. These were controlled for size on Photoshop CS and averaged for luminance using Matlab 2018a with the ShineToolbox 1.2 package [41]. Each image group was presented for 6 s, and looking durations were measured. To control for potential matching between image groups (presence of teeth in both faces and skulls), all faces depicted neutral expressions. Stones were geometrically manipulated into the outline of skulls. Due to the similarity of the stimuli, to avoid image fatigue, face and skull conditions were presented on different days together with the stone condition with a total of eight trials (one trial per day). We chose these three additional species since they are fairly common but also fairly distinct from each other. Apart from conspecific stimuli, the subjects were not overly familiar with the animal species presented, though they have seen both live and dead rodents (mice) and, to a lesser extent, live cats and dogs. None of them had any previous experience with skulls of any type. Each image group was presented for 6 s and was followed by a fixation cross at the middle. On each stimulus image (i.e. skull/face/stone), we drew areas of interest (AOIs) for each of the four species. These AOIs were slightly larger (10–15%) than the original stimuli to account for possible fixation errors.

For experiment 2, we presented only the chimpanzee stimuli used in the previous experiment with all three types (face, skull, stone), generating a total of 45 images to the chimpanzees (1 species × 3 types ×

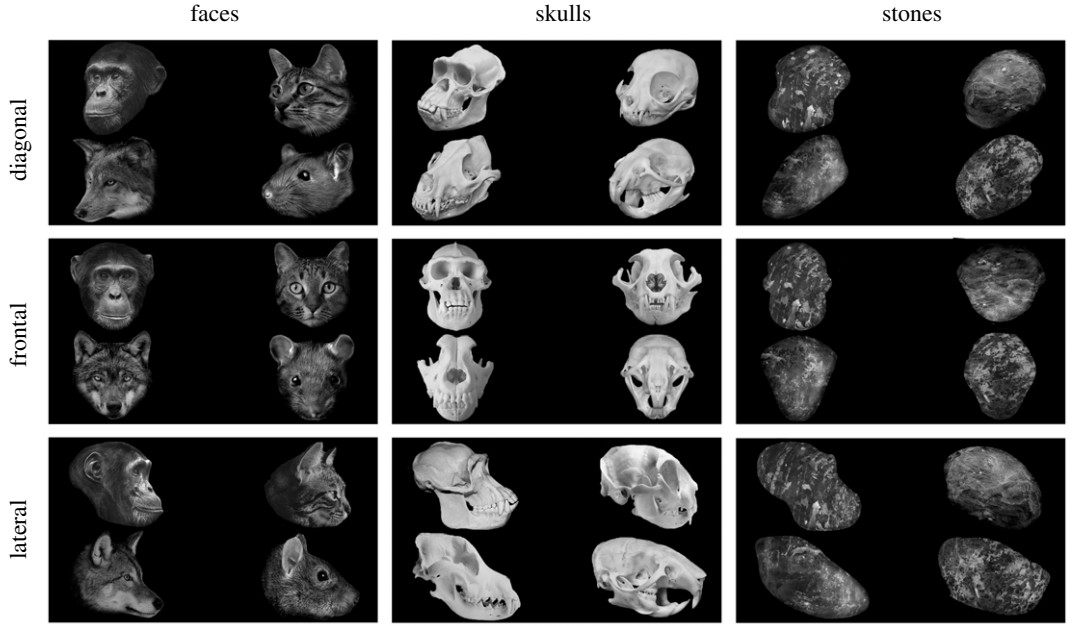

**Figure 3.** Examples of each image group stimuli presented to the chimpanzees (actual locations were randomized to avoid positional bias).

3 angles × 5 image variations). Each image group was presented again for 6 s for a total of five separate sessions. Three AOIs were likewise drawn around the image stimuli as with experiment 1. Each image group was presented for 6 s and followed by a fixation cross at the middle.

For experiment 3, we presented only the chimpanzee skulls used in experiments 1 and 2 at the frontal and diagonal orientations, a total of 10 images. Because chimpanzees have forward-facing eyes, this perceptual feature is lost when both faces and skulls are presented sideways on, we did not include the lateral orientations in this experiment. Three AOIs were drawn around the eye socket, the nasal cavity and the teeth regions. Each image was presented for 5 s and followed by a fixation cross randomly at each corner of the screen. For all experiments, eye movement data were filtered using a Tobii fixation filter. As our looking time measure, we used the total fixation duration measurement generated by the Tobii Studio software. Total fixation duration is the sum of the duration of all fixations (in seconds) occurring during stimulus presentation (figure 4 for an overview of the experimental flow).

## 2.4. Statistical analysis

For the first experiment (conspecific stimuli bias), we presented five variations of image group and then averaged the total fixation data by species accordingly (i.e. frontal skulls × 5 sessions, frontal faces × 5 sessions, frontal stones × 5 sessions). To detect any viewing preferences, we conducted a general linear mixed model (GLMM). We conducted three independent GLMM tests for each orientation (frontal, diagonal and lateral). Looking durations for stone were only counted once for the first viewing sessions. For the analyses, we set 'species' (chimp, cat, dog, rat) and 'type' (skull, face, stone) as within-subject factors and 'gaze' (total fixation duration) as a dependent variable with 'subject' (chimpanzee participants) nested in 'trial' as random factors.

For the second experiment (conspecific face bias), we presented all chimpanzee stimuli simultaneously at each of the three orientations (i.e. frontal face + frontal skull + frontal stone) for five sessions. A GLMM was also conducted with all three orientations pooled with 'gaze' as dependent variable, 'type' as within-subject factor and 'subject' nested in 'trial' as random factors.

For the third experiment (conspecific teeth bias), to determine where chimpanzees fixated their gaze in chimpanzee skulls, we presented chimpanzee skull images from both the frontal and diagonal orientations, one image at a time at the centre of the screen for five sessions. A GLMM was conducted with the frontal and diagonal orientations pooled, 'gaze' as dependent variable, 'type' (eye socket, nasal and teeth regions) as within-subject factors and 'subject' nested in 'trial' as random factors.

Since our data were both zero-inflated and continuous, all statistical analyses were conducted on the package glmmTMB [42] with the ziGamma family and the log link function (full model results in

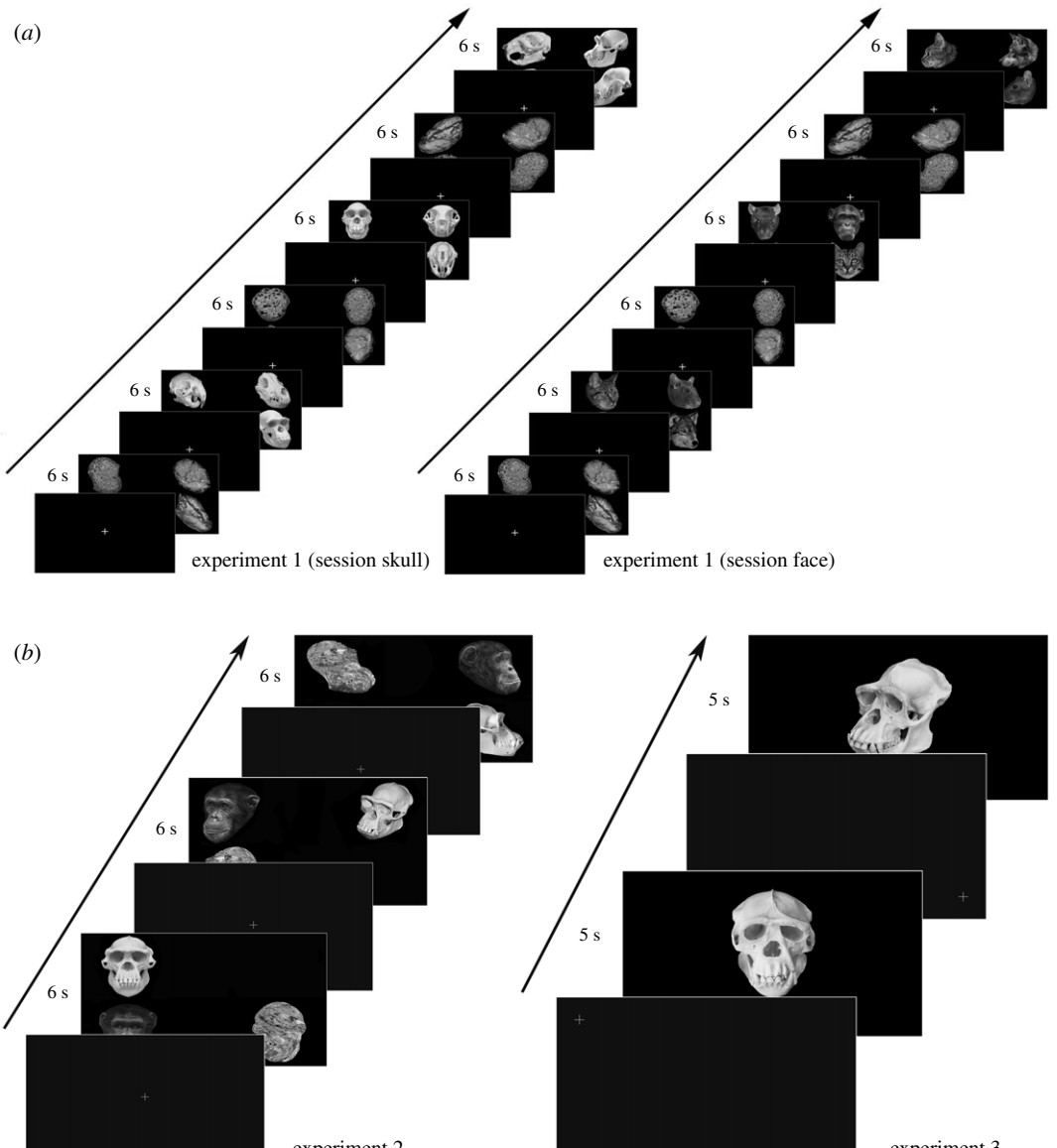

**Figure 4.** Experimental design flow for all three experiments.

electronic supplementary material, Data). The DHARMa package [43] was used to assess for inflation and dispersion. Significance for whole model terms was assessed with the 'drop1' function. *Post hoc* pairwise Bonferroni-corrected comparisons among levels (species and/or type) were conducted using the emmeans package. All analyses were conducted on R 4.1.0 [44].

# 3. Results

## 3.1. Experiment 1: conspecific versus heterospecific stimuli (faces, skulls, stones)

### 3.1.1. Frontal condition

In the frontal condition (observations, $N = 420$), there was a significant interaction between species and type ($\chi^2 = 14.39$, d.f. = 6, $p = 0.025$). In the skull sub-condition, chimpanzees looked longer overall at the *chimpanzee skulls* than for other species skulls. The following *post hoc* tests revealed this difference in fixation durations to be significant: *chimp/cat* ($t = -3.81$, $p = 0.0009$), *chimp/dog* ($t = 3.64$, $p = 0.0018$) and *chimp/rat* ($t = 4.48$, $p = 0.0001$). For the face sub-condition, fixations were again longer for the chimpanzee stimuli than for other species, and these differences were significant: *chimp/cat* ($t = -5.871$,

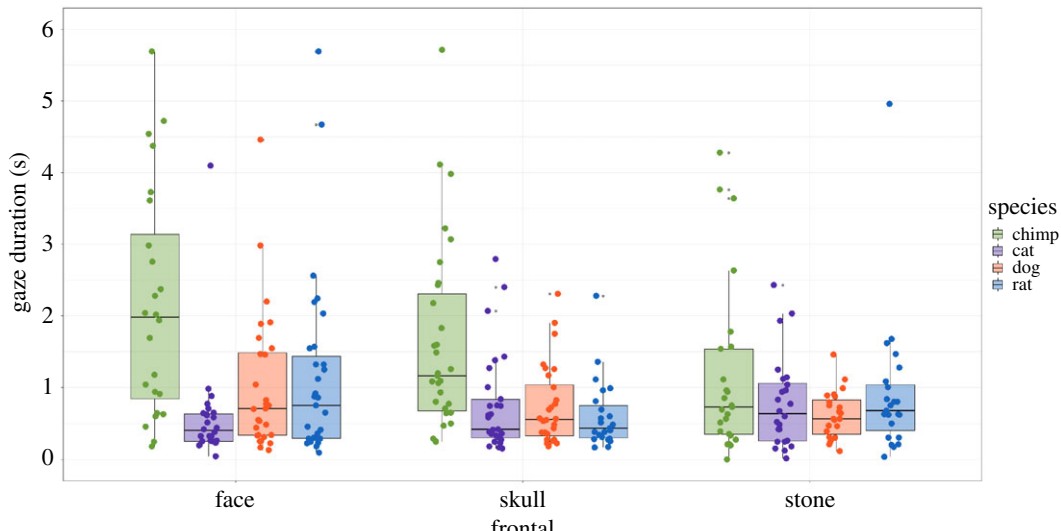

**Figure 5.** Looking duration values for the frontal condition.

**Table 1.** Fixation duration results per species and type for the frontal condition. Significant results shown in bold.

| type | species | gaze (s) | s.e. | contrasts | β | t-ratio | p-value | eff. size | 95% CI |
|------|---------|----------|------|-----------|-----|---------|---------|-----------|--------|
| skull | chimp | 1.559 | 0.166 | chimp–cat | −0.764 | −3.818 | **0.0009** | 1.05 | −1.63, −0.48 |
|       | cat | 0.726 | 0.170 | chimp–dog | 0.741 | 3.647 | **0.0018** | 1.08 | 0.51, 1.66 |
|       | dog | 0.743 | 0.172 | chimp–rat | 0.966 | 4.484 | **0.0001** | 1.33 | 0.76, 1.91 |
|       | rat | 0.593 | 0.186 | | | | | | |
| face | chimp | 2.174 | 0.185 | chimp–cat | −1.309 | −5.871 | **<0.0001** | 1.34 | −1.92, −0.76 |
|      | cat | 0.587 | 0.178 | chimp–dog | 0.755 | 3.435 | **0.0039** | 0.84 | 0.27, 1.41 |
|      | dog | 1.021 | 0.172 | chimp–rat | 0.698 | 3.220 | **0.0083** | 0.59 | 0.02, 1.16 |
|      | rat | 1.082 | 0.168 | | | | | | |
| stone | chimp | 1.180 | 0.184 | chimp–cat | −0.459 | −2.025 | 0.261 | 0.37 | −0.94, 0.19 |
|       | cat | 0.745 | 0.183 | chimp–dog | 0.672 | 2.964 | **0.019** | 0.54 | −0.02, 1.11 |
|       | dog | 0.602 | 0.185 | chimp–rat | 0.235 | 1.034 | 1.000 | 0.28 | −0.28, 0.85 |
|       | rat | 0.932 | 0.185 | | | | | | |

$p < 0.0001$), *chimp/dog* ($t = 3.43$, $p = 0.0039$) and *chimp/rat* ($t = 3.22$, $p = 0.0083$). Finally, for the stone sub-condition, looking durations were longer for the chimpanzee stimuli in comparison with other species, but these differences were non-significant, except for the dog: *chimp/cat* ($t = −2.02$, $p = 0.261$), *chimp/dog* ($t = 2.96$, $p = 0.019$) and *chimp/rat* ($t = 1.03$, $p = 1$) (table 1 and figure 5).

### 3.1.2. Diagonal condition

In the diagonal condition (observations, $N = 420$), we again found a significant interaction between species and type ($\chi^2 = 50.21$, d.f. = 6, $p < 0.0001$). In the skull sub-condition, while chimpanzees looked longer at the *chimpanzee skull* overall, the *post hoc* tests revealed that it was the comparison with the *dog skull* that drove the differences in fixation durations, with this difference significant: *chimp/cat* ($t = −0.37$, $p = 1$), *chimp/dog* ($t = 3.26$, $p = 0.007$) and *chimp/rat* ($t = 1.42$, $p = 0.931$). In the face sub-condition, the *post hoc* tests revealed that fixations were higher for the *chimp face* versus other species, with all differences significant: *chimp/cat* ($t = −6.92$, $p < 0.0001$); *chimp/dog* ($t = 5.39$, $p < 0.0001$) and *chimp/rat* ($t = 5.56$, $p < 0.0001$). Lastly, in the stone sub-condition, the *post hoc* tests revealed no significant effects among species, excepting the cat which had longer looking durations compared with the chimpanzee: *chimp/cat* ($t = 2.93$, $p = 0.021$), *chimp/dog* ($t = −1.24$, $p = 1$) and *chimp/rat* ($t = 1.03$, $p = 1$) (table 2 and figure 6).

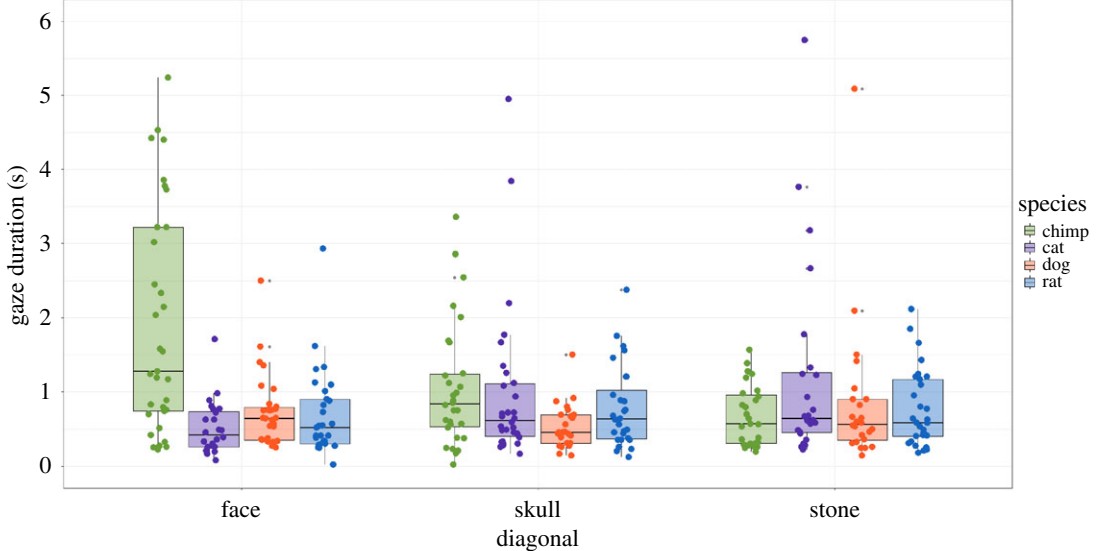

**Figure 6.** Looking duration values for the diagonal condition.

**Table 2.** Fixation duration results per species and type for the diagonal condition. Significant results shown in bold.

| type | species | gaze (s) | s.e. | contrasts | $\beta$ | $t$-ratio | $p$-value | eff. size | 95% CI |
|------|---------|----------|------|-----------|---------|-----------|-----------|-----------|--------|
| skull | chimp | 1.039 | 0.140 | chimp–cat | −0.069 | −0.371 | 1.000 | 0.09 | −0.60, 0.41 |
| | cat | 0.969 | 0.141 | chimp–dog | 0.644 | 3.266 | **0.007** | 0.90 | 0.35, 1.44 |
| | dog | 0.545 | 0.155 | chimp–rat | 0.269 | 1.424 | 0.931 | 0.37 | −0.14, 0.89 |
| | rat | 0.793 | 0.144 | | | | | | |
| face | chimp | 1.926 | 0.134 | chimp–cat | −1.337 | −6.924 | **<0.0001** | 1.87 | −2.40, −1.33 |
| | cat | 0.506 | 0.154 | chimp–dog | 0.983 | 5.391 | **<0.0001** | 1.37 | 0.87, 1.87 |
| | dog | 0.720 | 0.139 | chimp–rat | 1.027 | 5.569 | **<0.0001** | 1.43 | 0.92, 1.94 |
| | rat | 0.689 | 0.142 | | | | | | |
| stone | chimp | 0.670 | 0.156 | chimp–cat | 0.595 | 2.938 | **0.021** | 0.83 | 0.27, 1.38 |
| | cat | 1.215 | 0.156 | chimp–dog | −0.252 | −1.244 | 1.000 | 0.35 | −0.91, 0.20 |
| | dog | 0.862 | 0.156 | chimp–rat | −0.136 | 1.034 | 1.000 | 0.19 | −0.71, 0.33 |
| | rat | 0.768 | 0.141 | | | | | | |

### 3.1.3. Lateral condition

In the lateral condition (observations, $N = 420$), we found, yet again, significant interactions between species and type ($\chi^2 = 14.02$, d.f. = 6, $p = 0.0293$). The following *post hoc* tests in the skull sub-condition revealed no significant differences in fixation durations where *chimpanzee skulls* were compared: *chimp/cat* ($t = −1.46$, $p = 0.856$), *chimp/dog* ($t = 1.94$, $p = 0.311$) and *chimp/rat* ($t = −1.48$, $p = 0.823$). Unexpectedly, chimpanzees looked longer at *rat skulls* overall and significantly so when compared with *cat* and *dog skulls*. To further explore this effect, a follow-up lateral test was conducted with the rat skull's diastema covered. The previous significant results were no longer found in the follow-up test, although there was an overall higher trend for *chimpanzee skulls* (see §4.1 and electronic supplementary material, Data). In the face sub-condition, fixations were longer for the chimpanzee stimuli than for other species, and these differences were significant when compared with *cat* and *dog skulls*: *chimp/cat* ($t = −2.69$, $p = 0.044$), *chimp/dog* ($t = 2.26$, $p = 0.143$) and *chimp/rat* ($t = 2.78$, $p = 0.034$). For the stone sub-condition, looking durations were slightly longer for the chimpanzee stimuli in comparison with other species; however, these differences were non-significant: *chimp/cat* ($t = −0.85$, $p = 1$), *chimp/dog* ($t = 2.01$, $p = 0.268$) and *chimp/rat* ($t = 1.24$, $p = 1$) (table 3 and figure 7).

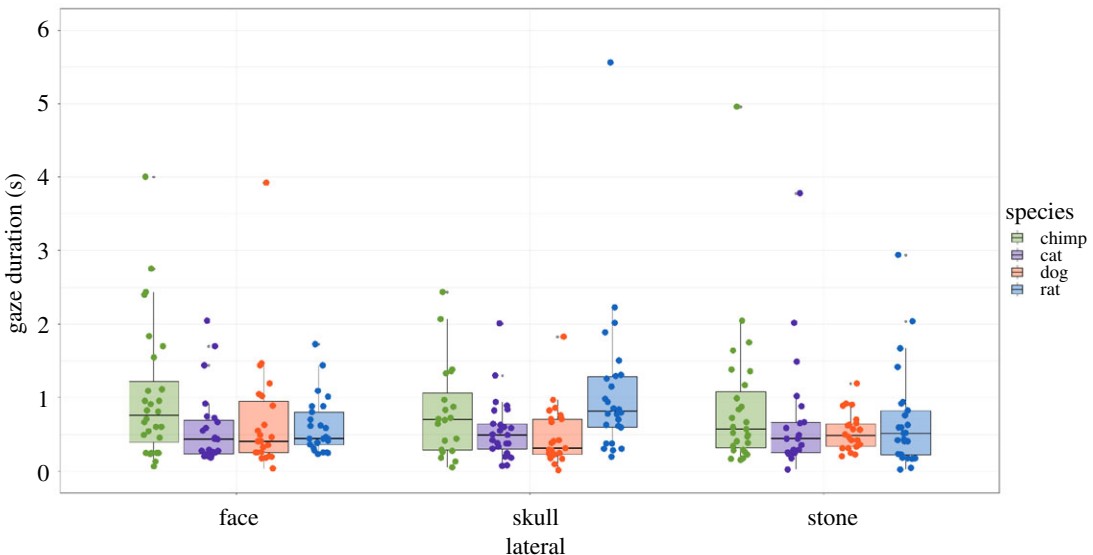

**Figure 7.** Looking duration values for the lateral condition.

**Table 3.** Fixation duration results per species and type for the lateral condition. Significant results shown in bold.

| type | species | gaze (s) | s.e. | contrasts | β | t-ratio | p-value | eff. size | 95% CI |
|------|---------|----------|------|-----------|---|---------|---------|-----------|--------|
| skull | chimp | 0.766 | 0.185 | chimp–cat | −0.340 | −1.469 | 0.856 | 0.44 | −1.04, 0.15 |
| | cat | 0.545 | 0.167 | chimp–dog | 0.463 | 1.949 | 0.311 | 0.60 | −0.006, 1.22 |
| | dog | 0.482 | 0.176 | chimp–rat | −0.343 | −1.489 | 0.823 | 0.45 | −1.04, 0.14 |
| | rat | 1.080 | 0.164 | | | | | | |
| face | chimp | 1.063 | 0.161 | chimp–cat | −0.591 | −2.696 | **0.044** | 0.77 | −1.34, −0.20 |
| | cat | 0.588 | 0.177 | chimp–dog | 0.505 | 2.267 | 0.143 | 0.66 | 0.08, 1.24 |
| | dog | 0.641 | 0.177 | chimp–rat | 0.593 | 2.780 | **0.034** | 0.78 | 0.22, 1.33 |
| | rat | 0.587 | 0.166 | | | | | | |
| stone | chimp | 0.852 | 0.162 | chimp–cat | −0.194 | −0.854 | 1.000 | 0.25 | −0.84, 0.33 |
| | cat | 0.701 | 0.182 | chimp–dog | 0.448 | 2.013 | 0.268 | 0.58 | 0.01, 1.16 |
| | dog | 0.544 | 0.176 | chimp–rat | 0.268 | 1.241 | 1.000 | 0.35 | −0.20, 0.91 |
| | rat | 0.651 | 0.169 | | | | | | |

## 3.2. Experiment 2: conspecific stimuli (faces with skulls and stones)

For experiment 2 (observations, $N = 315$) with orientations now pooled, there was a significant effect among chimpanzee types (face, skull, stone) ($\chi^2 = 39.11$, d.f. = 2, $p < 0.0001$). The results show that *chimpanzee faces* had the highest looking durations, followed by *chimpanzee skulls* and lastly *chimpanzee-shaped stones*. The following *post hoc* tests revealed that fixation durations were significantly longer for the *chimpanzee face* in comparison with *skull* and *stone* types, but not when *skull* and *stone* were compared with each other: *face/skull* ($t = 4.04$, $p = 0.0002$), *face/stone* ($t = 6.34$, $p < 0.0001$) and *skull/stone* ($t = 2.26$, $p = 0.073$) (table 4 and figure 8).

## 3.3. Experiment 3: conspecific skull regions (eye, nasal and teeth regions)

For experiment 3 (observations, $N = 210$), we found a significant effect among the chimpanzee skull AOIs (eye, nasal and teeth regions) ($\chi^2 = 24.73$, d.f. = 2, $p < 0.0001$). The *post hoc* tests revealed the longer fixations for *teeth* compared with the *eye* and *nasal* regions to be significantly different, whereas the relationship between eyes and nose was non-significant: *eyes/teeth* ($t = -4.25$, $p = 0.0001$), *nose/teeth* ($t = -4.42$, $p < 0.0001$) and *eyes/nose* ($t = 0.20$, $p = 1$) (table 5 and figure 9).

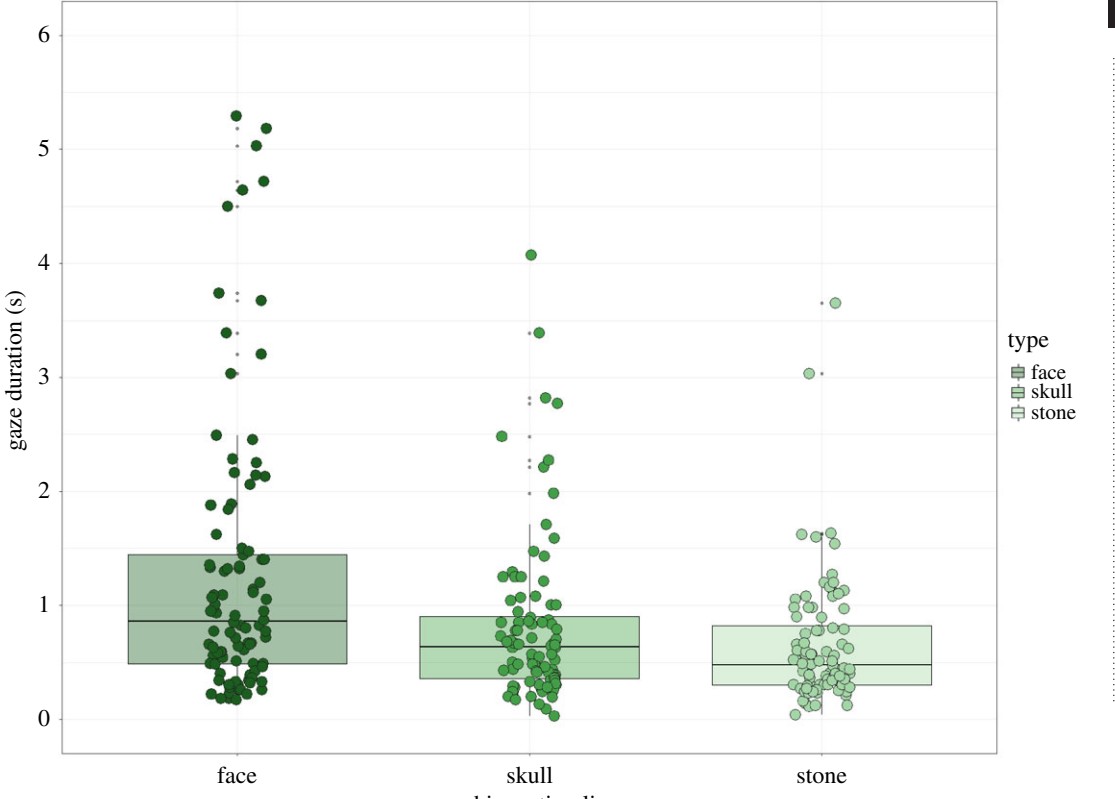

**Figure 8.** Looking duration values for the chimpanzee-only stimuli.

**Table 4.** Fixation duration results for the chimpanzee stimuli. Significant results shown in bold.

| type | areas | gaze (s) | s.e. | contrasts | $\beta$ | $t$-ratio | $p$-value | eff. size | 95% CI |
|------|-------|----------|------|-----------|---------|-----------|-----------|-----------|--------|
| chimp stimuli | face | 1.216 | 0.120 | face–skull | 0.428 | 4.048 | **0.0002** | 0.61 | −0.07, 0.39 |
| | skull | 0.707 | 0.122 | face–stone | 0.664 | 6.348 | **<0.0001** | 0.96 | −0.50, −0.02 |
| | stone | 0.564 | 0.121 | skull–stone | 0.237 | 2.263 | 0.073 | 0.34 | −0.74, −0.26 |

## 4. Discussion

### 4.1. General findings

Our primary aim was to find out if chimpanzees treat skulls similarly to faces (presumably via activation of a face module) and draw attention to the implications for comparative thanatology research where corpses in advanced states of decay and skeletons are involved. In the first experiment, with hypothesis 1a we wanted to determine if chimpanzees were relatively more interested in conspecific stimuli and with hypothesis 1b to see if such interest would also have an orientation effect, with a preference for frontal and diagonal over lateral orientations expected. In the second experiment, with hypothesis 2, we sought to ascertain whether chimpanzee faces were the driving factor guiding our subject's attention when these were placed with chimpanzee skulls and stones. Finally, with hypothesis 3, we needed to uncover whether such attention was concentrated on cues naturally visible in faces, in this case, teeth. The results show that they do but in a restricted sense. Overall, our subjects showed the most interest in conspecific faces, followed by conspecific skulls, and the least interest in conspecific skull-shaped stones. Moreover, this was particularly evident in frontal/diagonal orientations for chimpanzee faces and skulls. Furthermore, chimpanzees showed interest in conspecific teeth only relative to the eye and nasal regions of the skull (table 6).

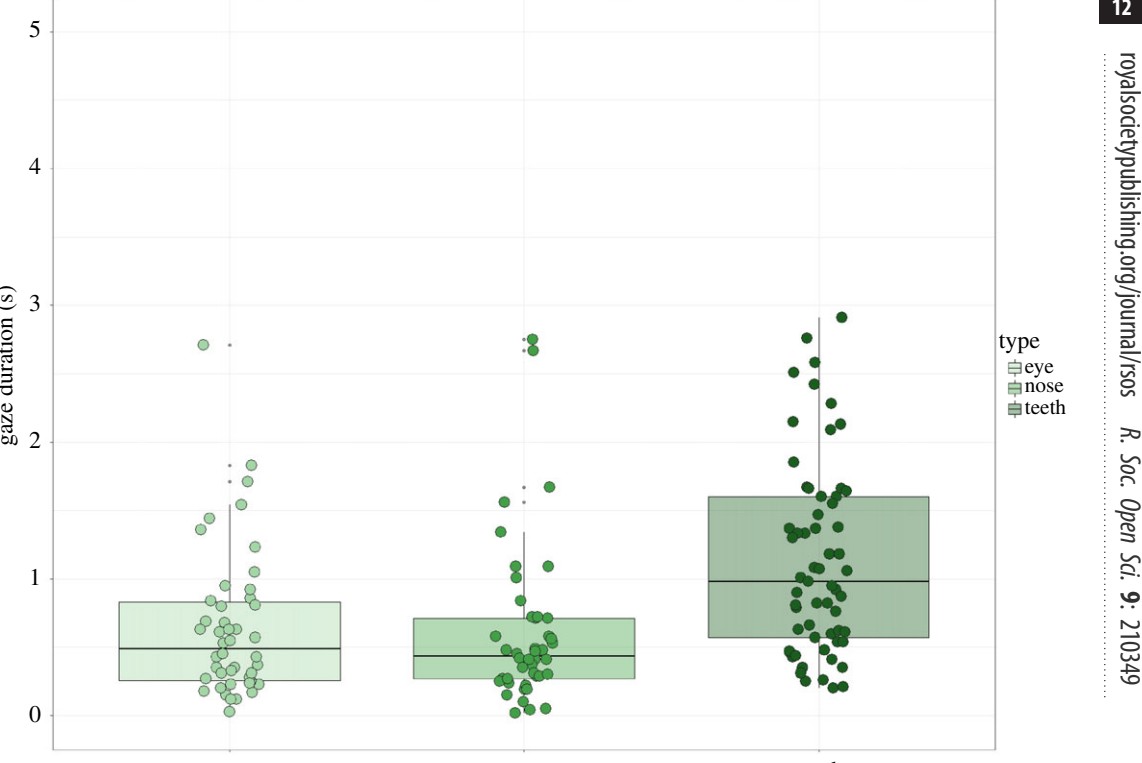

**Figure 9.** Looking duration values for the chimpanzee regions of the skull.

**Table 5.** Fixation duration results for the chimpanzee skull regions. Significant results shown in bold.

| type | areas | gaze (s) | s.e. | contrasts | $\beta$ | t-ratio | p-value | eff. size | 95% CI |
|---|---|---|---|---|---|---|---|---|---|
| chimp skull | eye | 0.607 | 0.140 | eye–nasal | 0.031 | 0.209 | 1.000 | 0.04 | −0.37, 0.46 |
| | nasal | 0.589 | 0.142 | eye–teeth | −0.596 | −4.259 | **0.0001** | 0.85 | −1.25, −0.45 |
| | teeth | 1.103 | 0.128 | nasal–teeth | −0.627 | −4.421 | **<0.0001** | 0.89 | −1.30, −0.49 |

Not only was this general interest in faces greater towards conspecific faces than for other species, but the same pattern was demonstrated for conspecific skulls and to a lesser extent for 'conspecific' stones, indicating that the main factor driving their interest was faces, in particular conspecific face-like stimuli. Moreover, the conditional models for all three conditions of experiment 1 show significant effects for *species chimp* (all types pooled) when compared with all three other *species* (all types pooled) (see electronic supplementary material, Data). What this suggests is that chimpanzees can extract familiar, face-like features from skulls (which retain multiple face-like features) although less so from objects (stones) only resembling faces/skulls in outline. Moreover, compared with frontal/diagonal orientations, the effects were weaker for lateral orientations than for frontal and diagonal orientations across all stimuli, presumably because lateral orientations carry less information about faces.

For experiment 1, in the frontal condition, chimpanzees showed a clear interest in frontally presented conspecific skulls; significant differences in looking times were noted for the chimpanzee skull versus all other non-conspecific skulls in this orientation. They also showed a high degree of interest in frontal conspecific faces, with their attention significantly greatest for chimpanzee faces relative to the *dog*, *cat* and *rat faces*. For the frontally oriented stones, chimpanzees again looked longer at the conspecific-like stones compared with the non-conspecific exemplars, but the looking durations were lower overall and the differences were non-significant, except when compared with *dog-shaped stones*. This suggests the stones, showing only the outlines of each species, were a too degraded facial stimulus type to retain their interest; a trend also shown for stones presented in other orientations.

**Table 6.** Research hypotheses and results.

| hypotheses | prediction | results | empirical support |
|---|---|---|---|
| H1a. conspecific stimuli bias | increased looking durations towards conspecific compared with non-conspecific stimuli. | confirmed | Heron-Delaney et al. [45]; Simpson et al. [46] |
| H1b. conspecific frontal bias | increased looking durations towards frontal/diagonal conspecific stimuli and decreased towards lateral conspecific stimuli. | confirmed | Burton and Bindemann [57]; Tomonaga and Imura [47] |
| H2. conspecific face bias | increased looking durations towards conspecific face followed by conspecific skulls and conspecific skull-shaped stones. | partially confirmed | this study |
| H3. conspecific teeth bias | increased looking durations for conspecific skulls towards the teeth compared with the eye and nasal regions. | confirmed | McComb et al. [12] |

In the diagonal condition for skulls, while chimpanzees exhibited the longest looking durations towards *chimpanzee skulls*, the difference was only significant when compared with *dog skulls*. Looking times were also of relatively long duration for the *cat skulls*, one possible explanation being that the feline skull has neotenic features at this orientation (a round skull and large orbits show a passing resemblance to an infant chimpanzee or monkey skull), and future research could address this. For diagonally presented faces, chimpanzees again showed a clear interest—significant differences in looking duration were noted for the chimpanzee face versus all three non-conspecific face types. These looking times for faces were also slightly higher in comparison with skulls. For diagonal stones, looking durations were yet lower overall and lowest for the *chimpanzee-shaped stones*, suggesting the stimuli lost much of their semblance to a chimpanzee face at this orientation. No significant effects were found, except for *cat-shaped stones* (longer) compared with *chimpanzee-shaped stones* (shorter).

In the lateral condition, fixations were decreased overall in comparison with diagonal and frontal conditions, and there were no significant differences among stimuli (except for *chimpanzee faces* compared with *cat* and *rat* faces), supporting the general idea that lateral orientations carry the least amount of facial/face-like information. The means were higher for chimpanzee stimuli across all types; however, no significant interaction was found between species and type except for the face sub-condition. As previously mentioned, the conditional models did yield significant results for species alone: the pooled stimuli (skull + face + stone) revealed that chimpanzees looked significantly longer at laterally presented chimpanzee stimuli compared with other species stimuli in the same orientation, the same was true for frontal and diagonal orientations (see electronic supplementary materials). One puzzling aspect of the lateral skull presentation was that chimpanzees seemingly attended more to the rat skull than to any other skull. We suspect this was due to the superficially open-mouth appearance (as if vocalizing); lateral orientations of rodent skulls make their diastema (space between front-facing teeth and remaining teeth) prominently visible. A lower level explanation is that the diastema makes the rat skull look perceptually salient causing it to stand out from all the other three. In an additional test, when we covered this area with bone texture, the chimpanzees no longer looked longer at the rat skulls relative to other non-conspecific species, but neither did the differences in their looking times reach significance for any other skull (see electronic supplementary materials).

In experiment 2, chimpanzees were presented with *chimpanzee faces* together with *chimpanzee skulls* and *chimpanzee-shaped stones*, in frontal, diagonal and lateral orientations. The combined results show that chimpanzees looked overall longer at *chimpanzee faces* followed by *chimpanzee skulls* and then *chimpanzee-shaped stones*. While the results were significant for *chimpanzee faces* compared with *skulls* and *stones*, the latter two (skulls and stones) when compared with each other were not significant, although these were close to significance, suggesting that a larger sample size might reveal a significant difference between the two (longer durations for skulls followed by stones).

In experiment 3, the specific areas within the *chimpanzee skulls* at both frontal and diagonal orientations, chimpanzees looked longer at the teeth, followed by the nasal area, and least towards the eye regions. The looking patterns were significantly longer for both teeth versus eye sockets and also for teeth versus nose cavities. These findings support both McComb *et al.*'s [12] study, in which elephants exhibited a greater interest in elephant tusks, and Watts' [26] prediction that chimpanzee skulls may represent iconic features of chimpanzee faces, notably in having teeth.

In conclusion, the most parsimonious explanation for these results is that chimpanzee skulls, essentially, exhibit a degraded signal of a face. Albeit impoverished, skulls nevertheless retain face-like features and appear to be subjected to the same facial biases. Presumably, the *relational properties* that linger in the skull (facial outline, teeth, nose cavity, eye sockets) activate a domain-specific face detection module that directs chimpanzees' attention to these approximated facial features.

## 4.2. Framing the results within face perception research

As in the present paper, *own-species facial bias* has been extensively documented in the human and non-human primate literature seemingly developing through *perceptual narrowing* (i.e. distinct areas of the brain become attuned to the faces individuals are exposed to most throughout development) (reviewed in [48]). While most of these studies employed methodologies different from our own, three are worth mentioning, as the simultaneous presentation of competing stimuli mirror ours. Research in humans using preferential looking found that while neonates did show a preference for human faces over non-human primate faces, they did not make such a distinction when whole bodies were presented. By contrast, 3.5- and six-month-olds showed a preference for human faces and human bodies when these were presented alongside non-human primates (rhesus monkeys or gorillas) [45]. These findings were reproduced in a similar study in which human and monkey faces were used; three-month-olds showed a clear preference for human faces, particularly the eyes [49]. Lastly, a preferential looking study involving rhesus monkey subjects found that three-week-old individuals were already better at locating face stimuli over non-face stimuli, and by three months of age, they attended to conspecific faces over non-conspecific faces [46].

Our research also builds upon previous findings using eye-tracking. Kano and Tomonaga [50], using naturalistic images of humans, chimpanzees and non-primate mammals, found that both humans and chimpanzees fixated first on faces, but that chimpanzees shifted their gaze more quickly. Similar results were found with gorillas and orangutans [51]. Furthermore, chimpanzees tended to fixate longer towards eyes, mouth and nose, respectively [50], while bonobos tended to look more at the eyes compared with chimpanzees, though both species look longer overall towards the eyes versus the mouth [52]. Strikingly, Hirata *et al.* [53] showed that although chimpanzees look more at the eye region of conspecific faces, when the eyes were closed they looked longer at the nose region followed closely by the mouth. This was mirrored, to some extent, in our study, given that the eye region of skulls (not having eyes) received lesser attention compared with teeth.

Much like our findings, other experiments also suggest processing differences between facial stimuli presented frontally and in profile. In a visual search task using touchscreens, Tomonaga and Imura [47] found that while chimpanzees were efficient at searching for forward-facing faces (human and chimpanzee), they were significantly slower at detecting faces shown in profile. This is echoed by previous facial recognition research on humans which consistently found poorer performance (slower reaction + lower accuracy) in trials using faces in profile compared with frontally facing [54–56]. One face detection study using visual search in natural scenes found that participants' performance was equally good for frontal and diagonally presented faces, but declined for faces in profile; this effect persisted even when only the upper halves of faces (eyes + nose bridge) were visible, but not for the lower portion (nostrils + mouth) [57]. Similar findings were also confirmed by Bindemann & Lewis [58]. Moreover, an eye-tracking study revealed that when human faces were viewed in profile, the perceived intensity of the facial expression was reduced compared with frontal and diagonal conditions [59]. Presumably, this is connected with direct gaze. According to Senju & Hasegawa [60], a direct gaze signals the valence of intention towards the receiver (i.e. communicative, affective, hostile, friendly or sexual) making it adaptive to direct attention towards front-on faces. This ability follows a developmental trajectory that starts with front-on faces being processed in dedicated areas of the brain, and somewhere around eight months of life, side-on faces are integrated into the same brain regions [61]. This is perhaps unsurprising since faces and bodies in profile are also involved in more elaborated computations such as decoding social cues (intentions/actions) towards third parties or objects (*sensu* [62]).

Neuroimaging research in humans has shown three distinct regions in the brain to activate when facial stimuli are presented. Within this dedicated network for face selectivity, the *fusiform gyrus* is considered its most robust and selective component [63], followed by the *superior temporal sulcus* and the *lateral occipital cortex* [30]. In comparison, macaques (*Macaca mulatta*), while lacking a clear hemispheric specialization and a *fusiform gyrus*, do show face selectivity in the *superior temporal sulcus* [64] and the *inferotemporal cortex* [65]. However, research in vervets (*Chlorocebus aethiops*) identifies a hemispheric asymmetry in favour of a stronger neuronal activation in the right *inferotemporal cortex* [66]. Furthermore, marmosets (*Callithrix jacchus*) show activation in the *superior temporal sulcus* as well as the *occipital cortex* [67]. In chimpanzees, face perception is not visibly lateralized as in humans, but activation of the *fusiform gyrus* as well as the *superior temporal sulcus* and the *orbitofrontal cortex* to face stimuli supports the claim that their facial processing is similar to humans [68]. Taken together, these studies raise the possibility that, while the *fusiform gyrus* specialization emerged recently in evolution and developed in humans and other apes, there is still a deep homology for face selectivity in specific brain areas of both Old World and New World primates.

Our findings are also in line with research in *face pareidolia* (detection of illusory faces on non-living objects) carried out in humans [69], chimpanzees [70] and rhesus monkeys [71] where inanimate face-like artefacts are perceived as faces. Neuroimaging analyses in humans situate illusory faces in the *fusiform gyrus* (specifically the *right fusiform face area*) [69,72] and indicate that, over time, this initial face detection migrates into brain areas dedicated towards object processing, suggesting recategorization from animate to inanimate [73]. The chimpanzee skull, we argue, falls within this class of highly pareidolic objects.

Thus, whereas the relationship between the face and the skull is not incidental (skulls functionally support faces and protect the brain), the attention towards forward-facing skulls is. This attention towards chimpanzee skulls is best explained as the by-product of a facial module (i.e. a network of specialized brain regions) originally evolved for processing facial features and emotional expressions.

## 4.3. Ecological considerations

There are some considerations regarding chimpanzee/elephant ecologies. First, most chimpanzees live in rainforest environments not conducive to the long-term preservation of bones. Woodland-savannah sites (i.e. Fongoli, Senegal or Issa Valley, Tanzania) might be better candidates for observing chimpanzee–skeleton interactions. But living in rainforests should not impact shorter term interactions. Forest elephants do live in these environments and have been known to interact with the carcases and bones of conspecifics. Two recent papers describe such visitations using camera traps; Stephan *et al.* [74] recorded 193 visits over eight months, and Hawley *et al.* [8] recorded five visits for three months. Chimpanzee remains, however, are typically collected for later pathological/anatomical/biomedical analysis (at some field sites though not all) [75], eliminating any potential interactions chimpanzees might have with them and therefore probably accounting for the paucity of observations [26]. For instance, a taphonomic study in Kibale recovered the remains of nine chimpanzees including the skulls (five complete with jawbones), while the remaining skeletal parts (ribs, fore and hind limbs) were either entirely missing or incomplete [76]. Dying on the forest floor might further accelerate scavenging, Yamagiwa [77] found the partially complete skeleton of an adult male chimpanzee in a one-month-old tree nest with other newer nests in the vicinity.

Second, unlike reported behaviour in elephants, chimpanzees show curiosity to all sorts of similar-sized dead animal species (i.e. aardvark, bushpig, bushbuck, leopard, monkeys) [2,27]. Chimpanzees are probably aware of conspecific skeletons in their environments, and these encounters are severely under-reported. This knowledge also extends to situations where mothers carry dead infants to the point of mummification/skeletonization [78]. Prey species with similar anatomy such as monkeys are also potentially informative; Boesch & Boesch [79] report on a juvenile chimpanzee with a colobus monkey skull using a tool to scoop out the brain, and another case involved an adult female using sticks to clean the eye sockets of a colobus skull after she had finished eating the eyes. Moreover, in some parts of Africa, chimpanzees are sympatric with lowland gorillas (for lethal encounters, see [80]), and gorilla skeletons, exceedingly similar to chimpanzees, are probably encountered. Naturally, being the largest extant land mammal, on dying, elephants leave the largest skulls in their habitats (roughly twice as large as a hippopotamus or a rhinoceros), the only other skulls that bear a superficial resemblance to theirs are those of their closest living relatives, the sirenians (dugongs and manatees) which, being strictly aquatic species, do not share the same habitats.

Finally, elephant tusks and chimpanzee teeth are tied to different contexts. Chimpanzee skulls bear a resemblance to particular facial expressions such as the fear grin and pout face. Likewise, chimpanzees' teeth become visible during screams and also during yawning or feeding. On the other hand, elephant tusks, always visible, can assume a more neutral character. Furthermore, without trunks and ears (two critical elements for communication), elephant skulls little resemble an actual elephant face. There remains the issue of whether chimpanzees in the wild associate skeletons with the places where dead individuals were previously seen. Just as with McComb *et al.*'s [12] study, our experiment demonstrates a general propensity in chimpanzees to look at same-species skulls which appears to be based perceptually, but this does not address directly whether they categorize skeletons or skulls as part of a dead conspecific, a question which would benefit from further exploration. Regarding to what extent and how chimpanzees might behave towards skeletons in the wild, in light of our findings, two, not necessarily mutually exclusive, explanations can be advanced: chimpanzees pay attention to conspecific chimpanzee skeletons (particularly to the skull as a by-product of face-processing skills), and they may infer this meaning (and in some cases the identity of the individual) from the places in which they are found. Such questions may also have a bearing on research into the *landscapes of fear and disgust*, investigating whether dead bodies provide information on danger due to predation or pathogens (*sensu* [81]).

# 5. Conclusion and future directions

We began this study with the central assumption that chimpanzee skulls are perceived like degraded chimpanzee faces and that they would likewise be subjected to the same biases. We proposed three working hypotheses: H1a, chimpanzees look longer at conspecific stimuli versus non-conspecific stimuli (conspecific stimuli > non-conspecific stimuli); H1b, chimpanzees look longer at frontal/diagonal conspecific stimuli versus laterally presented conspecific stimuli (frontal ≈ diagonal > lateral); H2, within conspecific stimuli, chimpanzees look longer at *chimpanzee faces* followed by *skulls* and *stones* (face > skull > stone) and H3, just as elephants direct their attention towards elephant tusks, likewise chimpanzees look longer at conspecific teeth versus other facial regions (teeth > eye ≈ nose). Overall, we found support for all three hypotheses. For H1a, chimpanzees exhibited significantly longer looking durations towards conspecific relative to non-conspecific stimuli when types were pooled (see electronic supplementary material, Data). They also looked significantly longer across most types (skull and face) and orientations (frontal and diagonal) except for stone stimuli (looking durations were relatively longer toward frontal *chimpanzee stones*, but the difference was only significant when compared for dog stones) reinforcing the 'degraded face assumption'. For H1b, chimpanzees showed significantly longer looking durations for frontal/diagonal conspecific stimuli in comparison with laterally presented conspecific stimuli, again showing similar biases to previous facial research experiments. For H2, with the chimpanzee-only stimuli, the chimpanzees did look significantly longer at the *chimpanzee faces* compared with *chimpanzee skulls* and *chimpanzee-shaped stones*, but this dropped below significance when comparing the *chimpanzee skull* with the *chimpanzee-shaped stone*, although the direction of difference fitted our prediction further supporting the 'degraded face assumption'. For H3, in the chimpanzee skull regions, our prediction that chimpanzees would look predominantly at the teeth compared with other areas was also upheld. They looked significantly longer at the teeth versus the eye socket and the nasal regions of the skull.

The combined results show support for our hypotheses and do suggest a connection between a domain-specific module in the chimpanzee brain directing their attention towards face-like stimuli. This face module evolved and develops within the context of face-to-face interactions (the likely reason all frontal conditions in our experiment, the chimpanzee stimuli received longer looking patterns overall). The skull contains relevant, albeit impoverished face-like features. This relationship is, of course, not incidental, as skulls support faces, but the attention towards skulls appears to be best explained as a by-product of a module originally evolved for decoding facial expressions. Perhaps notably, unlike wild chimpanzees, our captive subjects never interacted with conspecific skeletons. This suggests that, apart from learned associations, similar interest exhibited by their wild counterparts towards conspecific skulls might also be explained by the same recognition mechanism. To further decode the phylogeny of this face–skull relationship, future studies could compare naive human infants' performance in a similar task (1–3-year-olds familiar with human faces, but with no experience of human skulls). Another research avenue would be to replicate McComb *et al.*'s [12] experiment in the laboratory with the aid of a three-dimensional printer (skulls controlled for size and

colour). Finally, neuroimaging studies could further address the precise connection between skull and face stimuli in the brain. The question put forth by Christophe Boesch, at the beginning of our paper, pondering what goes on in the chimpanzee's mind when they encounter conspecific skulls remains unanswerable. In the light of this study, our tentative answer must also, in the end, be phrased as a question: strange, yet familiar?

Ethics. All procedures adhered to institutional guidelines (PRI: Primate Research Institute 2002 version of 'The Guidelines for the Care and Use of Laboratory Primates'). The experimental design was approved by the Animal Welfare and Animal Care Committee of the PRI (2020–228, 2021-152) and the Animal Research Committee of Kyoto University.

Data accessibility. Supporting data are provided in the electronic supplementary material [82].

Authors' contributions. A.G.: conceptualization, formal analysis, investigation, methodology, writing—original draft, writing—review and editing; Y.H.: writing—review and editing; I.A.: supervision, writing—review and editing.

All authors gave final approval for publication and agreed to be held accountable for the work performed therein.

Competing interests. We declare we have no competing interests.

Funding. This research was supported by JSPS (Japanese Society for the Promotion of Science) KAKENHI (grant no. 19J15133).

Acknowledgements. We thank M. Tomonaga for his continued support and valuable comments. We thank E. Ichino along with the other members of the Language and Intelligence Section for their assistance and comments in conducting the experiments. We also thank Y. Kim, C. Sarabian and J. Gao for their critical insights and C.F.I. Watson for the English editing. Finally, the authors would like to extend their gratitude to the five anonymous reviewers whose comments significantly improved the final version of this manuscript.

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
