## [Peer Review File · Royal Society Open Science]

Review History

RSOS-210349.R0 (Original submission)

Review form: Reviewer 1

Is the manuscript scientifically sound in its present form?

Yes

Are the interpretations and conclusions justified by the results?

Yes

Is the language acceptable?

Yes

Do you have any ethical concerns with this paper?

No

Have you any concerns about statistical analyses in this paper?

Yes

Recommendation?

Accept with minor revision (please list in comments)

Comments to the Author(s)

This is a really important, fascinating and understudied topic, which the authors should be commended for pursuing. The stimuli categories are nicely controlled. I have only minor clarification comments.

Why were the particular species of skulls chosen? How were the stones created/selected?

Within the main text, the presentation of stimuli needs to be more detailed. How were the stimuli presented – in pairs, one at a time? How many per session? How balanced per order? What did the chimpanzees do on each trial? How many sessions were presented to each subject? Why did three of the subjects drop out?

The structure for the analysis is unclear. Why was AOI one factor – why not species and material (skull, stone)? What are the interactions in the model?

The results are similarly not written up in a clear manner. The authors need to refer explicitly to significant main effects and interactions and specify the factors more clearly.

How can there be a significant effect between the DV and AOIS?? (line 53)

Why analyze the faces, skulls and stones separately? Was there an interaction that called for splitting the analysis this way?

It would have been interesting to see whether there was the same conspecific preference for complete skeletons except for skulls, which would negate the 'face attention' explanation.

Why the preference for rat skulls in the lateral condition? I may have missed it but I didn't think the authors addressed this in the discussion.

P. 9, line 8, I don't follow the reference to second order relational info

Line 32 "where" should be "were."

Line 59 missing 'not to?'

Review form: Reviewer 2 (David Watts)

Is the manuscript scientifically sound in its present form?

Yes

Are the interpretations and conclusions justified by the results?

Yes

Is the language acceptable?

No

Do you have any ethical concerns with this paper?

No

Have you any concerns about statistical analyses in this paper?

Yes

Recommendation?

Major revision is needed (please make suggestions in comments)

Comments to the Author(s)

This manuscript describes an interesting study that contributes to the small literature on how chimpanzees respond to faces and the smaller literature on how they respond to skeletal

elements, especially skulls. I think the authors could substantially improve the work, though, and I will offer some criticisms and suggestion that I think can help them to do this.

First, I should say that I have great sympathy for colleagues who are not native English speakers, but who are required to write in English for most academic journals in their fields. I could not write publishable quality work in Portuguese or Japanese (or in any language other than English). However, the manuscript could be substantially shorter and many points would be clearer if the authors have someone who is a native English-speaker help them with copy editing. Also, eliminating some of the extra words and the repetition would make the work more concise and easier for readers to understand. For example, the last sentence of the Abstract contains five phrases set off by commas that precede a final statement. Much of the wording is redundant and the authors could omit it. Having a single, short sentence that makes a clear statement would be much better. This same logic applies to many other places in the manuscript. Also, the authors should use past tense in reporting their results and then in describing them in the Discussion. This is partly they are reporting on events that occurred in the past. Importantly, also is because when we make statements like “chimpanzees do X”, instead of explicitly saying that “in this study, the chimpanzees did X”, we seem to be saying that the results apply to all chimpanzees, everywhere and at all times. I have made some copy-editing suggestions below.

More substantively, I would like to see the authors give more attention to the literature on face recognition. The paper is not just about responses to skulls. The authors state in the Discussion (P. 7, line 59) that one of their goals was to assess whether attention to skulls and head-like shapes was “guided by underlying interest in conspecific faces”. Some of the work on “interest in faces” has involved investigating how well primates (including humans) can discriminate among individual faces, which was not a question the authors were trying to address. However, I think they would benefit by looking at Taubert’s work (*Perception*, 38: 343, 2009) on “holistic” processing by humans of non-conspecific faces (including those of chimpanzees. She found evidence that humans process chimpanzee faces holistically – as we do faces of conspecifics – but we seem not to do so for spider monkeys or gorillas. I think this has some relevance to the question of whether chimpanzees process skulls, face-shaped (or really head-shaped) objects, or heterospecific faces like they do chimpanzee faces. Likewise, Taubert et al. (2017, *Animal Cognition* 20:321) found evidence that chimpanzees, like humans, are better attuned to averaged facial images than to specific ones; I suspect that this also has some implications for the authors’ work. In a series of experiments that also used eye-tracking, Kano et al. (2018, *PLoS One*) found that chimpanzees presented with video images looked longer at the mouths than the eyes of conspecifics and of other primate species (so did humans), the opposite of the result from the present study. I realize that the stimuli were different, which might explain the different results. However, I think the authors need to use this paper in contextualizing their study and their results. They could also benefit by noting that Kano et al. found considerable inter-individual variation (although less for chimpanzees than for the other species in their study), some of it related to variation in rearing histories, and by echoing Kano et al.’s emphasis on the fact that many primates attend selectively to conspecific eyes and faces, follow gaze, etc. In fact, we have a big literature on the importance of faces in primate social worlds. (Incidentally, Kano et al. found that bonobos looked longer at eyes than faces; they suggested that this difference from chimpanzees might have occurred because bonobos are more tolerant of eye contact.) I suggest also that the authors look at the methods Kano et al. used, which differed from their own; actually, the general research question was different, because Kano et al. were interested in differences among chimpanzees, bonobos, rhesus, orangutans, and humans, but I think that paper offers valuable information for comparison. Kano and Tomanaga (2010; *Animal Behavior* 79:227) also deserves attention.

My sense is that the lack of attention to studies like these was because the authors chose to limit their comparisons mostly to the studies of elephants that they cite. Those comparisons are

interesting, but I don't think they tell us much that is new about chimpanzees. In turn, this is partly because the authors list "evolutionary thanatology" as one of the manuscript's keywords, but they have not really focused on what their results add to the literature on evolution thanatology other than to note that their results showed that their subjects gave more attention to chimpanzee skulls than to those of other species and that their particular attention to the teeth area is consistent with the idea that this skull region most resembles the faces of living chimpanzees. I would like the authors to say more about what this means for evolutionary thanatology, especially with reference to Goncalves and Carvalho (2019), which I think is one of the best contributions to this literature. Doing that in the Discussion instead of repeating results would make the paper more valuable.

I also think the paper would benefit from a table that summarizes the hypotheses and predictions and then notes which predictions were supported. This would make it easier for readers to follow the presentations of all the statistical results.

Finally, I would like to see some discussion of the average overall times that the chimpanzees looked at faces, skulls, or stones. I realize that they did not design the study to investigate whether the chimpanzees looked longer at chimpanzee faces than at chimpanzee skulls or stones shaped like chimpanzee faces/heads. We might expect them to look longest at faces, although the face-like aspects of skulls combined with their novelty might have undermined such an effect. Perhaps they could use their data to investigate this issue, although trying to do a statistical analysis may not be valid. In any case, I noticed the following: mean looking time was shorter for the stone than the face or skull in all conditions; it was longer for the face than the skull in the diagonal condition, but the absolute difference was quite small, and it was approximately equal for the face and skull in the frontal condition. I think this deserves some comment. Also, the subjects looked longer at chimpanzee faces than rat faces in only one of three conditions. Was this perhaps because of their experience seeing live rats?

One general comment on methods and reporting of results: I have not yet been convinced that we should abandon p-values. However, if we use conventional statistics and report these values, we should not report "trends". Trends are non-significant, by definition, and we should report the results as "not significant".

Specific comments (page numbers refer to pages in upper right of manuscript, e.g. 2 of 12):

P. 3, first full sentence: Omit "a fairly obscure paper" and just say "Halder & Schenkel (1972) found that, like elephants, bovids [which species?] responded more to bones of conspecifics than to those of other species."

P. 3, line 5: Saying "without any benefit" isn't necessary. How would elephants benefit by interacting with bones of conspecifics?

P. 3, last sentence of 2.1: First, say that these authors presented bones to the elephants (and indicate that these were elephants in the wild), not that they presented elephants to the bones. Also, this sentence could be shorter and clearer. "Whereupon" should be "on which". Make a second sentence that is simply "The elephants showed most interest in elephant bones", then a final sentence "Rasmussen (ref.) obtained similar results for captive elephants."

P. 3, 2.2: Omit the first sentence and just say "Two classic studies provided insights into chimpanzee responses to skeletons."

P. 3, line 51: This should be "in chimpanzees and gorillas"; "among" makes it sound like gorillas and chimpanzees were interacting with each other.

2.3: This paragraph would benefit from some editing to make it shorter and clearer (especially lines 11-13).

P. 5, line 6, "together with stone": Your description of "image groups" and the figure depicting examples of these indicates that a group consisted of either four images of skulls, 4 images that each showed a different species, or four images of skull-shaped stones. "Together with stone" makes it sound like you showed images of stones at the same time as you showed images of faces or skulls. Please make it clear that you showed the stone image groups on every day of testing, but alternated face groups and skull groups.

P. 5, line 8: Omit "significant".

P. 5, line 15: Missing "were" after "AOIs".

P. 5, 37-39: Omit the material about "trends". Alternatively, the most you can say is that the mean values differed, but not significantly.

P. 5, line 42: Make it clear what was "diminished".

P. 5, line 42, sentence, starting "Since": Omit this sentence; you already said this in the Intro.

P. 5, line 43, sentence starting "For analyses": this should be in Methods.

P. 5, line 45: "are" should be "were".

P. 5, line 46: "who" should be "which".

P. 5, line 53: Either say "there was either say there was "a significant relationship" between these variables or that "AOI significantly affected gaze duration".

P. 5, line 54: Here and in other places where you present results of statistical tests (especially Tukey results), it would better to write "Overall fixation durations were longer [better word than "higher"] for the chimpanzee skull than for all others (cat: $t = -4.18$, $p = 0.003$, Cohen's $d=0.83$; dog: $t = 4.13$, $p < 0.001$, Cohen's $d=0.84$ ", etc. You already stated that you used Tukey HSD tests; you don't need to repeat that for each comparison. Also, you can "t" instead of "t-ratio" and don't need to repeat "chimp/rat", etc. when you start by indicating that you are presenting the results of comparisons between chimps (o chimp-like stones) and each of the other stimuli.

P. 6, line 5: "positively non-significant" should be "positive, but non-significant".

4.1.2 and later presentation of statistical results: as I urged above, please re-write this to say that you found (or did not find) significant relationships between variables, then state the results of the Tukey HSD tests as I suggested for p. 5, line 54.

P. 6, line 19: Omit "once again".

P. 6, line 20: Please do not invoke "marginal" effects. The relationship was non-significant.

P. 6, line 38, sentence starting "Because": This should be in Methods. Also, you don't need to repeat that you ran an ANOVA for the skull regions (line 40); just give the result.

Discussion:

The first paragraph would benefit from shortening and some re-organization. This also applies to several of the later sections. The entire Discussion could be considerably shorter and more concise. For one thing, you don't need to repeat so many of the results (a table would help; see my comment above).

P. 6, line 60, "visible face cues": how could they orient to non-visible cues? I think you mean "cues that are visible on faces".

P. 7, lines 1-3: This needs some re-writing. Again, don't invoke "trends". Also, the stones were not "conspicifics"; you mean stones that resembled the heads of conspecifics (or heterospecifics).

P. 7, line 4: re-write as "from skulls, which retain multiple face like features, but not objects that resemble only face outlines".

P7, line 6: "is" should be "was", and I suggest you re-write this as "The effects were weaker for lateral orientations than for frontal and diagonal orientations for all stimuli, presumably because lateral orientations carry less information about faces". (Skulls & stones do not carry information about facial expressions!)

P 7, line 12, "higher looking times": Better as "Looking times were longer for the cat skull..."

P. 7, line 25, sentence starting "Finally": "even more decreased" should be "even shorter" and "among" should be "for".

P. 7, next sentence: "this higher looking tendency among" should be "the longer looking time for".

P. 7, paragraph starting 34: I can understand your point, but please see if you can make it more clearly.

Section 5.2: The material on a proposed "face module" could use some up-dating.

Section 5.3: Not all chimpanzees live in rainforests (line 19).

P. 8, lines 25-27: Skeletal remains are only collected at research sites, and not at all of them. Also, researchers do not find the carcasses of most deceased chimpanzees, and I suspect that chimpanzees are much more likely to encounter them than researchers are. Also, chimpanzees in the wild have experience handling both the faces (heads) and skulls of species on which they prey; monkeys comprise most prey, and their faces and skulls certainly resemble those of chimpanzees.

P. 8, line 34: Teeth are also visible when chimpanzees yawn (very common!), screams, or fight, and they are often visible while individuals are feeding.

Section 6: You don't need to repeat the results. Just give a summary of what you think your results mean.

Review form: Reviewer 3

Is the manuscript scientifically sound in its present form?

Yes

Are the interpretations and conclusions justified by the results?

Yes

Is the language acceptable?

Yes

Do you have any ethical concerns with this paper?

No

Have you any concerns about statistical analyses in this paper?

No

Recommendation?

Major revision is needed (please make suggestions in comments)

Comments to the Author(s)

The manuscript entitled “Staring death in the face: Chimpanzees’ attention towards conspecific skulls and the implications of a face-module guiding this behaviour” presents results from an investigation of how captive chimpanzees attend to images of conspecific skulls, compared to those of non-conspecifics. Having established the state of the art of how elephants respond to conspecific skulls, as well as pointing out a lack of comparable studies on chimpanzees, despite these two species sharing several socio-cognitive traits, the authors justify the rationale of their study as “we aimed to find out, if chimpanzees exhibited the similar interest patterns as observed with African elephants.” As their findings, the authors report that chimpanzees pay most attention to images of conspecific faces, followed by those of their skulls and skull-shaped stones. Moreover, chimps scan the conspecific faces and skulls from teeth-region upwards till the nose.

I have two major concerns and a minor one, regarding the theoretical premise of this study, as well as interpretation of its results and their implication.

a) In my opinion, it would be better if the authors invoke a theoretical rationale with respect to primate ethology and evolution to justify the choice of the subject of their study. Right now, the rationale sits heavily on elephant-studies. While it is a pertinent rationale on grounds of comparative cognitive research, I believe the manuscript could benefit from bringing in a theoretical framework explaining the adaptive value of an ability to recognise conspecific skulls in their natural environment, for chimpanzees in particular, and primates in general.

b) Do the findings of this particular study add anything novel to our existing knowledge on face-perception by chimpanzees? I do not make this remark after having seen the results, on the contrary, the study design itself made me wonder what different results could be expected than from eye-tracking studies on conspecific face recognition by chimpanzees. Are the skull-stimuli, used in this study was a proxy of the face-stimuli. Are the hypotheses a roundabout way to ask “Do chimpanzees look longer at conspecific-faces than non-conspecific ones?” the latter had already been addressed previously, therefore, the authors need to establish the novelty of this study, especially in comparison to extant facial recognition studies on chimpanzees.

I think the seed of potentially addressing this concern is already sown by the authors in this manuscript where they reflect, “Not only was this general interest in skulls greater for the conspecific skulls, but the same trend was shown slightly higher for conspecific faces but lesser for conspecific stones, which suggests the main factor for their attraction are faces and particularly conspecific face-like stimuli. What this suggests is that chimpanzees are able to extract familiar face-like features from distinct non-face stimuli such as skulls which still retain said features and fading in conspecific stones where only the outline of a face is shown.” This point should be the main pitch of this manuscript, addressing a crucial question, “when does the face-ness of conspecific faces cease to exist for chimpanzees?” In focussing on this particular

question, the manuscript potentially could address a very interesting problem in cognitive science, that of perception of essence of objects. The capacity to identify an object, to a limit, even when they have lost some of their component parts, also form building blocks of formation of concepts and has profound implications in larger fields of such as that of cognitive linguistics.

c) The third concern is an extension of the above one. It is important that the authors discuss their rationale and findings in the context of previous face-recognition studies, using eye-tracking technology, on chimpanzees. In section 5.2 they discuss about brain areas responsible for facial recognition in primates. However, what remains missing is a more relevant discussion relating to the studies by Fumihiro Kano and colleagues on eye tracking research on chimpanzee face recognition (more so as those studies were largely done in the same facility with common chimps as participants).

Below are my specific comments for respective sections. Since the manuscript did not have line numbers specified, I used the subheadings of the sections, thereafter quoting the particular lines to add my comments on them.

Section 2.3 Research motivation & questions: The authors mention, “Moreover, if such interest was guided by some sort of recognition, would chimpanzees exhibit similar looking times towards conspecific faces (hypothesis 2)?” : clarification necessary for the rather vague phrase, ‘guided by some sort of recognition’, especially while stating a hypothesis to be tested.

Section 3.3 Procedure & Stimuli: I understand that the skull-sizes were controlled for, something that the authors rectified after McComb et al 2006. However, my concern lies with dimensions of the non-conspecific skulls. For example, the characteristic formation of the frontal bone and the temporal arches of the non-conspecifics would be markedly different from those of the conspecifics. Therefore, the area of image covered by the stimuli would not be controlled for each species. This particular concern about the area of stimuli presented is somewhat reflected in Fig 2(c) skull condition where the rat-skull takes up more longitudinal area than the others. Fig 3 confirms this, where we see that looking duration on the rodent skull in lateral condition was indeed significantly higher than on the rest.

Now, to test hypothesis 1 of this study, the authors are failing their precondition: “With different sets of skulls, all else being the same...”, because apart from image size, all other factors are not being the same. On the other hand, does one need to control for the skull-size at all, if the goal is to test whether chimpanzees recognize their conspecific skulls? The difference in skull sizes between non-specifics could perhaps be controlled by including skulls of infant or juvenile chimpanzees into the stimuli-set.

My suggestions would be, the authors should justify, their decision to correct for size of the images and not dimensions of the skulls.

Section 4.1.4 Teeth & Mouth Regions: The authors find that chimpanzees looked longer at the teeth region of conspecific skulls. This could be an outcome of exposed teeth in fear-grimace facial expressions of chimpanzees, therefore, could be a confounding result. We are not sure if the chimpanzees are paying attention to a particular emotion, i.e. fear indicated by exposed teeth, or the teeth area of a skull.

Section 5.2 Framing the results within face processing research: As pointed out earlier, this sections need reflection on previous eye-tracking studies on face recognition by chimpanzees (e.g. by Kano et al), how finding of the present study relate to previous results, and how this manuscript goes beyond previous findings, thereby establishing its novelty.

Review form: Reviewer 4

Is the manuscript scientifically sound in its present form?

Yes

Are the interpretations and conclusions justified by the results?

Yes

Is the language acceptable?

No

Do you have any ethical concerns with this paper?

No

Have you any concerns about statistical analyses in this paper?

Yes

Recommendation?

Major revision is needed (please make suggestions in comments)

Comments to the Author(s)

The manuscript presents an interesting finding. It advances the literature on how chimpanzees process face-related information by presenting apes with different species' skulls, faces, and skull-shaped stones. The study also complements previous literature on how chimpanzees react to dead conspecifics. However, although the study presents the results in a very detailed manner, I propose an alternative way to analyze the results (below). Furthermore, the study misses many interesting references related to natural encounters of chimpanzees with dead conspecifics. Finally, the study needs to clarify other points before acceptance. Below, I provide detailed comments. Unfortunately, the article is not number lined; thus, I can only refer to the main sections to target the comments.

Summary

Break down the very last sentence for a better flow. As of now, it contains six commas, and it is a bit difficult to follow.

Introduction

The authors start their introduction by discussing both chimpanzee and elephant literature. Then suddenly, after the sentence starting "Even more,..." they only refer to elephants and continue referring to non-human animals in general. I would restructure this first part of the introduction, perhaps moving from general findings across taxa to specific chimpanzee-elephant comparisons. At the beginning of page 3, the authors cite Halder & Schenkel 1972 as a "fairly obscure paper". Either elaborate on that point or remove that adjective.

Chimpanzees

I am a bit confused with this section of the introduction. The author cites two classic experiments but misses the opportunity to cite much more recent and perhaps, relevant work by authors including van Leeuwen, Cronin or Biro to name a few. It seems that most of the literature on chimpanzees' natural encounters with dead bodies is missing. Authors should include crucial references, especially given the limited amount of literature in this field.

Cronin, K. A., Van Leeuwen, E. J., Mulenga, I. C., & Bodamer, M. D. (2011). Behavioral response of a chimpanzee mother toward her dead infant.

Van Leeuwen, E. J., Mulenga, I. C., Bodamer, M. D., & Cronin, K. A. (2016). Chimpanzees' responses to the dead body of a 9-year-old group member. *American Journal of Primatology*, 78(9), 914-922.

Subjects

The authors mention that there were three dropouts. Please state why this was the case.

Procedure and stimuli

The authors have counterbalanced very well all combinations of condition, sub-conditions, and species. However, I believe the paragraph needs clarification. At first sight, it is hard to find where the 144 trials come from. I would suggest the authors do something similar to what they are already doing in the Statistical Analysis section (e.g., information in brackets: 4 species X 3 conditions X 3 sub-conditions X 4 variations for a total of 144 trials).

Out of curiosity, I wonder if researchers considered including human faces to control for familiarity given how unfamiliar they were to all animals except conspecifics.

I found a bit unclear which conditions did they received on each testing day. For example, were face and skull conditions always presented together with stones? In other words, days with face and stone and days with skull and stone? I believe this is not the case, and every day they received either skulls or faces or stones, but I am not entirely sure.

Statistical Analysis

I was puzzled that the authors did not statistically analyze whether chimpanzees look significantly longer to faces vs. skulls vs. skull-shape stones. For instance, if chimpanzees' average looking time is not significantly longer for skulls than skull-shape stones, the significance of the overall results may change.

To do that, the authors could try LMM with a Gaussian error structure. A linear mixed model would allow authors to study in the same model the effects of species, condition (skull, face, skull-shaped stone) and sub-conditions together with any interaction between these variables. The authors could also control for any learning and order effects by including session and/or trial numbers. Such a model would also allow authors to not just control for subject ID as a random effect (as in simple ANOVAs) but also the random slopes between the random effect (subject) and the main effects (e.g. condition, species, sub-conditions). The package the authors use is suitable for the proposed analysis.

Notice, though, that I am not stating that the current analyses are incorrect. On the contrary, the authors could extract more information from their data while better controlling the effect of specific variables. Furthermore, the authors would reduce the amount of ANOVAs (eight if I am not mistaken) to just a couple of models focusing on overall looking times and on specific regions of interest within the chimpanzee sub-sample.

Results

Overall looking time

I am not entirely sure why the authors need the first paragraph of the result section. If this is a requisite of the journal, it is ok, but it feels like part of the discussion. It is anticipating the results that are later presenter in greater detail with statistical values.

This first section also needs some rephrasing in any case. The authors mention that there appears to be a significant trend across skull face and stone conditions regarding stimuli in frontal sub-conditions. This is right, but then I do not understand why right after this sentence, the authors reiterate that there was more substantial attention to diagonal and frontal (again) sub-conditions in both skull and face but diminished in the lateral sub-conditions. I think it is better to first present the effect on the frontal sub-condition at once.

The overall looking time section seems to introduce the results. I would clarify here why the authors only considered diagonal and frontal chimpanzee faces and skulls for investigating the stimuli chimpanzee look at. The authors only explain that by the end of the results section.

Skulls and faces

In these two results sections, the authors mention, by the end of both paragraphs, that "all other comparisons were non-significant". Do the authors refer to comparisons within the lateral sub-conditions or in general? The statement seems to be especially unclear for the face sections.

Teeth and mouth

I wondered what was the reason not to show animal faces (or at least chimpanzee faces for this analysis) with the mouth open so that teeth were visible? Was it due to the potential emotional reactions they elicit (e.g., fear grin face in chimpanzees)? I understand the authors cannot control for eye presence in skulls, but it is possible to show teeth in alive faces. It might be worth clarifying the reasons for future readers.

Discussion

General findings

The sentence starting with "Moreover, compared to..." is very unclear. I would recommend reformulating it. Similarly, the sentence discussing the non-significant difference between chimpanzees' and cats' frontal skulls is hard to follow.

The authors state that for the lateral skull sub-condition, there were no significant differences among stimuli. Please correct the statement. They seem to look longer for rats than all other species – and there are significant differences between rats and dogs.

The end of the second discussion paragraph needs references.

Further considerations

The end of this section is slightly confusing. It needs rephrasing (e.g., ..they previously seen dead..).

Conclusions

The sentence starting "For H1;.." needs rephrasing.

For H2, why is the non-significant trend between chimpanzees and rats' frontal faces likely to disappear with a larger sample size? Curious to know why this is the only trend interpreted that way.

Finally, by the end of the manuscript (page 9) the authors discuss that the hypothetical face-module develop within the context of frontal face-to-face interactions for the first time. Before, they only refer to this module in general terms. Could you elaborate why it evolved in the context of face-to-face interactions? As of now it reads as a conclusion stemming solely from your results.

Review form: Reviewer 5

Is the manuscript scientifically sound in its present form?

Yes

Are the interpretations and conclusions justified by the results?

No

Is the language acceptable?

No

Do you have any ethical concerns with this paper?

No

Have you any concerns about statistical analyses in this paper?

Yes

Recommendation?

Major revision is needed (please make suggestions in comments)

Comments to the Author(s)

The authors present a novel eye-tracking study in chimpanzees that examines visual attention to pictures of faces, skulls, and stones (that mimicked the shape/color of the species). Four different

species of mammals were presented: chimpanzee, cat, dog, and rat, and each condition was presented at three different angles (diagonal, frontal and lateral). As such, this is an impressive dataset for examining heterospecific and conspecific face perception in chimpanzees along three main lines of inquiry: 1) looking times to conspecific versus heterospecific faces (which is not in itself novel, but the inclusion of these particular mammalian species is), 2) whether looking times change with angle of presentation and if so, whether the changes are consistent across con/heterospecifics or not, 3) whether progressive ‘degrading’ of the face signal (from face to skull to stone) alters looking times, and whether this is consistent across con/heterospecifics, 4) what are the salient features of the presented cues. Unfortunately, rather than using the theoretical framework of face processing here, the authors attempted to ground their research in comparative thanatology, which is not the most appropriate framing here. Consequently, I do not think this manuscript is publishable in its current form. If the authors were willing to substantially revise the manuscript, including the theoretical framework, then I think it could be a publishable contribution.

The manuscript should also be edited for clarity/word choices throughout. There are some unnecessarily long and confusing sentences (e.g. the last sentence of 1. Summary).

Specific comments:

Introduction:

As described above, the thanatological framework is weak. The authors themselves note that it is extremely rare for chimpanzees to come across conspecific skulls, so I am a bit confused as to why they chose to justify this study in this manner. While this is worth some treatment in the discussion, there is a far more rich literature on how chimpanzees and other primates process faces that provide appropriate background and justification for the study including:

For comparing looking times to con- and heterospecific faces

- 1) Kano and Tomonaga 2009 - chimpanzee viewing patterns when shown pictures of chimpanzees, humans, other mammals
- 2) Hattori et al. 2010 - differential sensitivity to conspecific and allospecific cues in chimpanzee and humans

Angles of presentation/ degradation of signal:

- 1) Tomonana & Imura 2009
- 2) Dahl et al. 2013
- 3) Taubert et al. 2012
- 4) Herman et al. 1990 (in dolphins!)

Face scanning/salient features - many, but including

- 1) Hirata et al. 2010
- 2) Kano et al. 2015
- 3) Kano et al. 2012

Methods:

Please provide more description on calibration methods, I do not know what “two small clips on each referent point” means.

Figure 1. is quite confusing since the stimuli in front of the chimpanzee in the figure looks nothing like the examples provided in Figure 2, please clarify.

Figure 2. Please rework this to show the species in the same configuration across all conditions (e.g. chimpanzee top left, dog top right, cat bottom left, rat bottom right). I understand that this is not how they were presented to subjects (to avoid positional biases), but it would be useful to be able to match the stone condition directly to the species/angle of interest.

More information needs to be provided on the statistical methods. What does AOI (species-stimuli) refer to? Is this, for example, cat-face (vs cat-skull) or cat-frontal versus cat-lateral? It seems like it is the skull/face/stone category differences later in the paragraph, but then I don't understand how the angle of presentation was analyzed. Please clarify this so that your work is replicable.

Results:

Please provide descriptive statistics (mean looking times and sds) for all comparisons.

Please also refer to the appropriate figures in the text of the results.

A summary table of results would be much more digestible.

Discussion:

The discussion should be reworked according to the suggested framing above.

Decision letter (RSOS-210349.R0)

Dear Mr Goncalves

The Editors assigned to your paper RSOS-210349 "Staring death in the face: Chimpanzees' attention towards conspecific skulls and the implications of a face-module guiding this behavior" have now received comments from reviewers and would like you to revise the paper in accordance with the reviewer comments and any comments from the Editors. Please note this decision does not guarantee eventual acceptance.

Please submit your revised manuscript and required files (see below) no later than 21 days from today's (ie 06-Jul-2021) date. Note: the ScholarOne system will 'lock' if submission of the revision is attempted 21 or more days after the deadline. If you do not think you will be able to meet this deadline please contact the editorial office immediately.

Please note article processing charges apply to papers accepted for publication in Royal Society Open Science (<https://royalsocietypublishing.org/rsos/charges>). Charges will also apply to papers transferred to the journal from other Royal Society Publishing journals, as well as papers submitted as part of our collaboration with the Royal Society of Chemistry

(<https://royalsocietypublishing.org/rsos/chemistry>). Fee waivers are available but must be requested when you submit your revision (<https://royalsocietypublishing.org/rsos/waivers>).

on behalf of Professor Essi Viding (Subject Editor)
openscience@royalsociety.org

Associate Editor Comments to Author:

Please accept our apologies for the unusual delay in completing the review of your paper - after initially struggling to secure sufficient reviewers to complete reports, we unexpectedly received a surfeit of reports (as you'll see). Many of the reviewers offer helpful suggestions and have queries that should be addressed in your revision. Given the volume of feedback, which may be tricky to complete in our usual 21 day timeline for a revision, please let the editorial office know if you need a couple of weeks additional time to ensure you thoroughly address the queries raised.

Reviewer comments to Author:

Reviewer: 1

Comments to the Author(s)

This is a really important, fascinating and understudied topic, which the authors should be commended for pursuing. The stimuli categories are nicely controlled. I have only minor clarification comments.

Why were the particular species of skulls chosen? How were the stones created/selected?

Within the main text, the presentation of stimuli needs to be more detailed. How were the stimuli presented - in pairs, one at a time? How many per session? How balanced per order? What did the chimpanzees do on each trial? How many sessions were presented to each subject? Why did three of the subjects drop out?

The structure for the analysis is unclear. Why was AOI one factor - why not species and material (skull, stone)? What are the interactions in the model?

The results are similarly not written up in a clear manner. The authors need to refer explicitly to significant main effects and interactions and specify the factors more clearly.

How can there be a significant effect between the DV and AOIS?? (line 53)

Why analyze the faces, skulls and stones separately? Was there an interaction that called for splitting the analysis this way?

It would have been interesting to see whether there was the same conspecific preference for complete skeletons except for skulls, which would negate the 'face attention' explanation.

Why the preference for rat skulls in the lateral condition? I may have missed it but I didn't think the authors addressed this in the discussion.

P. 9, line 8, I don't follow the reference to second order relational info

Line 32 "where" should be "were."

Line 59 missing 'not to?'

Reviewer: 2

Comments to the Author(s)

This manuscript describes an interesting study that contributes to the small literature on how chimpanzees respond to faces and the smaller literature on how they respond to skeletal

elements, especially skulls. I think the authors could substantially improve the work, though, and I will offer some criticisms and suggestion that I think can help them to do this.

First, I should say that I have great sympathy for colleagues who are not native English speakers, but who are required to write in English for most academic journals in their fields. I could not write publishable quality work in Portuguese or Japanese (or in any language other than English). However, the manuscript could be substantially shorter and many points would be clearer if the authors have someone who is a native English-speaker help them with copy editing. Also, eliminating some of the extra words and the repetition would make the work more concise and easier for readers to understand. For example, the last sentence of the Abstract contains five phrases set off by commas that precede a final statement. Much of the wording is redundant and the authors could omit it. Having a single, short sentence that makes a clear statement would be much better. This same logic applies to many other places in the manuscript. Also, the authors should use past tense in reporting their results and then in describing them in the Discussion. This is partly they are reporting on events that occurred in the past. Importantly, also is because when we make statements like “chimpanzees do X”, instead of explicitly saying that “in this study, the chimpanzees did X”, we seem to be saying that the results apply to all chimpanzees, everywhere and at all times. I have made some copy-editing suggestions below.

More substantively, I would like to see the authors give more attention to the literature on face recognition. The paper is not just about responses to skulls. The authors state in the Discussion (P. 7, line 59) that one of their goals was to assess whether attention to skulls and head-like shapes was “guided by underlying interest in conspecific faces”. Some of the work on “interest in faces” has involved investigating how well primates (including humans) can discriminate among individual faces, which was not a question the authors were trying to address. However, I think they would benefit by looking at Taubert’s work (Perception, 38: 343, 2009) on “holistic” processing by humans of non-conspecific faces (including those of chimpanzees. She found evidence that humans process chimpanzee faces holistically – as we do faces of conspecifics – but we seem not to do so for spider monkeys or gorillas. I think this has some relevance to the question of whether chimpanzees process skulls, face-shaped (or really head-shaped) objects, or heterospecific faces like they do chimpanzee faces. Likewise, Taubert et al. (2017, Animal Cognition 20:321) found evidence that chimpanzees, like humans, are better attuned to averaged facial images than to specific ones; I suspect that this also has some implications for the authors’ work. In a series of experiments that also used eye-tracking, Kano et al. (2018, PLoS One) found that chimpanzees presented with video images looked longer at the mouths than the eyes of conspecifics and of other primate species (so did humans), the opposite of the result from the present study. I realize that the stimuli were different, which might explain the different results. However, I think the authors need to use this paper in contextualizing their study and their results. They could also benefit by noting that Kano et al. found considerable inter-individual variation (although less for chimpanzees than for the other species in their study), some of it related to variation in rearing histories, and by echoing Kano et al.’s emphasis on the fact that many primates attend selectively to conspecific eyes and faces, follow gaze, etc. In fact, we have a big literature on the importance of faces in primate social worlds. (Incidentally, Kano et al, found that bonobos looked longer at eyes than faces; they suggested that this difference from chimpanzees might have occurred because bonobos are more tolerant of eye contact.) I suggest also that the authors look at the methods Kano et al. used, which differed from their own; actually, the general research question was different, because Kano et al. were interested in differences among chimpanzees, bonobos, rhesus, orangutans, and humans, but I think that paper offers valuable information for comparison. Kano and Tomanaga (2010; Animal Behavior 79:227) also deserves attention.

My sense is that the lack of attention to studies like these was because the authors chose to limit their comparisons mostly to the studies of elephants that they cite. Those comparisons are

interesting, but I don't think they tell us much that is new about chimpanzees. In turn, this is partly because the authors list "evolutionary thanatology" as one of the manuscript's keywords, but they have not really focused on what their results add to the literature on evolution thanatology other than to note that their results showed that their subjects gave more attention to chimpanzee skulls than to those of other species and that their particular attention to the teeth area is consistent with the idea that this skull region most resembles the faces of living chimpanzees. I would like the authors to say more about what this means for evolutionary thanatology, especially with reference to Goncalves and Carvalho (2019), which I think is one of the best contributions to this literature. Doing that in the Discussion instead of repeating results would make the paper more valuable.

I also think the paper would benefit from a table that summarizes the hypotheses and predictions and then notes which predictions were supported. This would make it easier for readers to follow the presentations of all the statistical results.

Finally, I would like to see some discussion of the average overall times that the chimpanzees looked at faces, skulls, or stones. I realize that they did not design the study to investigate whether the chimpanzees looked longer at chimpanzee faces than at chimpanzee skulls or stones shaped like chimpanzee faces/heads. We might expect them to look longest at faces, although the face-like aspects of skulls combined with their novelty might have undermined such an effect. Perhaps they could use their data to investigate this issue, although trying to do a statistical analysis may not be valid. In any case, I noticed the following: mean looking time was shorter for the stone than the face or skull in all conditions; it was longer for the face than the skull in the diagonal condition, but the absolute difference was quite small, and it was approximately equal for the face and skull in the frontal condition. I think this deserves some comment. Also, the subjects looked longer at chimpanzee faces than rat faces in only one of three conditions. Was this perhaps because of their experience seeing live rats?

One general comment on methods and reporting of results: I have not yet been convinced that we should abandon p-values. However, if we use conventional statistics and report these values, we should not report "trends". Trends are non-significant, by definition, and we should report the results as "not significant".

Specific comments (page numbers refer to pages in upper right of manuscript, e.g. 2 of 12):

P. 3, first full sentence: Omit "a fairly obscure paper" and just say "Halder & Schenkel (1972) found that, like elephants, bovids [which species?] responded more to bones of conspecifics than to those of other species."

P. 3, line 5: Saying "without any benefit" isn't necessary. How would elephants benefit by interacting with bones of conspecifics?

P. 3, last sentence of 2.1: First, say that these authors presented bones to the elephants (and indicate that these were elephants in the wild), not that they presented elephants to the bones. Also, this sentence could be shorter and clearer. "Whereupon" should be "on which". Make a second sentence that is simply "The elephants showed most interest in elephant bones", then a final sentence "Rasmussen (ref.) obtained similar results for captive elephants."

P. 3, 2.2: Omit the first sentence and just say "Two classic studies provided insights into chimpanzee responses to skeletons."

P. 3, line 51: This should be "in chimpanzees and gorillas"; "among" makes it sound like gorillas and chimpanzees were interacting with each other.

2.3: This paragraph would benefit from some editing to make it shorter and clearer (especially lines 11-13).

P. 5, line 6, "together with stone": Your description of "image groups" and the figure depicting examples of these indicates that a group consisted of either four images of skulls, 4 images that each showed a different species, or four images of skull-shaped stones. "Together with stone" makes it sound like you showed images of stones at the same time as you showed images of faces or skulls. Please make it clear that you showed the stone image groups on every day of testing, but alternated face groups and skull groups.

P. 5, line 8: Omit "significant".

P. 5, line 15: Missing "were" after "AOIs".

P. 5, 37-39: Omit the material about "trends". Alternatively, the most you can say is that the mean values differed, but not significantly.

P. 5, line 42: Make it clear what was "diminished".

P. 5, line 42, sentence, starting "Since": Omit this sentence; you already said this in the Intro.

P. 5, line 43, sentence starting "For analyses": this should be in Methods.

P. 5, line 45: "are" should be "were".

P. 5, line 46: "who" should be "which".

P. 5, line 53: Either say "there was either say there was "a significant relationship" between these variables or that "AOI significantly affected gaze duration".

P. 5, line 54: Here and in other places where you present results of statistical tests (especially Tukey results), it would better to write "Overall fixation durations were longer [better word than "higher"] for the chimpanzee skull than for all others (cat: $t = -4.18$, $p = 0.003$, Cohen's $d=0.83$; dog: $t = 4.13$, $p < 0.001$, Cohen's $d=0.84$ ", etc. You already stated that you used Tukey HSD tests; you don't need to repeat that for each comparison. Also, you can "t" instead of "t-ratio" and don't need to repeat "chimp/rat", etc. when you start by indicating that you are presenting the results of comparisons between chimps (o chimp-like stones) and each of the other stimuli.

P. 6, line 5: "positively non-significant" should be "positive, but non-significant".

4.1.2 and later presentation of statistical results: as I urged above, please re-write this to say that you found (or did not find) significant relationships between variables, then state the results of the Tukey HSD tests as I suggested for p. 5, line 54.

P. 6, line 19: Omit "once again".

P. 6, line 20: Please do not invoke "marginal" effects. The relationship was non-significant.

P. 6, line 38, sentence starting "Because": This should be in Methods. Also, you don't need to repeat that you ran an ANOVA for the skull regions (line 40); just give the result.

Discussion:

The first paragraph would benefit from shortening and some re-organization. This also applies to several of the later sections. The entire Discussion could be considerably shorter and more concise. For one thing, you don't need to repeat so many of the results (a table would help; see my comment above).

P. 6, line 60, "visible face cues": how could they orient to non-visible cues? I think you mean "cues that are visible on faces".

P. 7, lines 1-3: This needs some re-writing. Again, don't invoke "trends". Also, the stones were not "conspicifics"; you mean stones that resembled the heads of conspecifics (or heterospecifics).

P. 7, line 4: re-write as "from skulls, which retain multiple face like features, but not objects that resemble only face outlines".

P7, line 6: "is" should be "was", and I suggest you re-write this as "The effects were weaker for lateral orientations than for frontal and diagonal orientations for all stimuli, presumably because lateral orientations carry less information about faces". (Skulls & stones do not carry information about facial expressions!)

P 7, line 12, "higher looking times": Better as "Looking times were longer for the cat skull..."

P. 7, line 25, sentence starting "Finally": "even more decreased" should be "even shorter" and "among" should be "for".

P. 7, next sentence: "this higher looking tendency among" should be "the longer looking time for".

P. 7, paragraph starting 34: I can understand your point, but please see if you can make it more clearly.

Section 5.2: The material on a proposed "face module" could use some up-dating.

Section 5.3: Not all chimpanzees live in rainforests (line 19).

P. 8, lines 25-27: Skeletal remains are only collected at research sites, and not at all of them. Also, researchers do not find the carcasses of most deceased chimpanzees, and I suspect that chimpanzees are much more likely to encounter them than researchers are. Also, chimpanzees in the wild have experience handling both the faces (heads) and skulls of species on which they prey; monkeys comprise most prey, and their faces and skulls certainly resemble those of chimpanzees.

P. 8, line 34: Teeth are also visible when chimpanzees yawn (very common!), screams, or fight, and they are often visible while individuals are feeding.

Section 6: You don't need to repeat the results. Just give a summary of what you think your results mean.

Reviewer: 3

Comments to the Author(s)

The manuscript entitled "Staring death in the face: Chimpanzees' attention towards conspecific skulls and the implications of a face-module guiding this behaviour" presents results from an investigation of how captive chimpanzees attend to images of conspecific skulls, compared to those of non-conspecifics. Having established the state of the art of how elephants respond to

conspecific skulls, as well as pointing out a lack of comparable studies on chimpanzees, despite these two species sharing several socio-cognitive traits, the authors justify the rationale of their study as “we aimed to find out, if chimpanzees exhibited the similar interest patterns as observed with African elephants.” As their findings, the authors report that chimpanzees pay most attention to images of conspecific faces, followed by those of their skulls and skull-shaped stones. Moreover, chimps scan the conspecific faces and skulls from teeth-region upwards till the nose.

I have two major concerns and a minor one, regarding the theoretical premise of this study, as well as interpretation of its results and their implication.

a) In my opinion, it would be better if the authors invoke a theoretical rationale with respect to primate ethology and evolution to justify the choice of the subject of their study. Right now, the rationale sits heavily on elephant-studies. While it is a pertinent rationale on grounds of comparative cognitive research, I believe the manuscript could benefit from bringing in a theoretical framework explaining the adaptive value of an ability to recognise conspecific skulls in their natural environment, for chimpanzees in particular, and primates in general.

b) Do the findings of this particular study add anything novel to our existing knowledge on face-perception by chimpanzees? I do not make this remark after having seen the results, on the contrary, the study design itself made me wonder what different results could be expected than from eye-tracking studies on conspecific face recognition by chimpanzees. Are the skull-stimuli, used in this study was a proxy of the face-stimuli. Are the hypotheses a roundabout way to ask “Do chimpanzees look longer at conspecific-faces than non-conspecific ones?” the latter had already been addressed previously, therefore, the authors need to establish the novelty of this study, especially in comparison to extant facial recognition studies on chimpanzees.

I think the seed of potentially addressing this concern is already sown by the authors in this manuscript where they reflect, “Not only was this general interest in skulls greater for the conspecific skulls, but the same trend was shown slightly higher for conspecific faces but lesser for conspecific stones, which suggests the main factor for their attraction are faces and particularly conspecific face-like stimuli. What this suggests is that chimpanzees are able to extract familiar face-like features from distinct non-face stimuli such as skulls which still retain said features and fading in conspecific stones where only the outline of a face is shown.” This point should be the main pitch of this manuscript, addressing a crucial question, “when does the face-ness of conspecific faces cease to exist for chimpanzees?” In focussing on this particular question, the manuscript potentially could address a very interesting problem in cognitive science, that of perception of essence of objects. The capacity to identify an object, to a limit, even when they have lost some of their component parts, also form building blocks of formation of concepts and has profound implications in larger fields of such as that of cognitive linguistics.

c) The third concern is an extension of the above one. It is important that the authors discuss their rationale and findings in the context of previous face-recognition studies, using eye-tracking technology, on chimpanzees. In section 5.2 they discuss about brain areas responsible for facial recognition in primates. However, what remains missing is a more relevant discussion relating to the studies by Fumihiro Kano and colleagues on eye tracking research on chimpanzee face recognition (more so as those studies were largely done in the same facility with common chimps as participants).

Below are my specific comments for respective sections. Since the manuscript did not have line numbers specified, I used the subheadings of the sections, thereafter quoting the particular lines to add my comments on them.

Section 2.3 Research motivation & questions: The authors mention, “Moreover, if such interest was guided by some sort of recognition, would chimpanzees exhibit similar looking times towards conspecific faces (hypothesis 2)?” : clarification necessary for the rather vague phrase, ‘guided by some sort of recognition’, especially while stating a hypothesis to be tested.

Section 3.3 Procedure & Stimuli: I understand that the skull-sizes were controlled for, something that the authors rectified after McComb et al 2006. However, my concern lies with dimensions of the non-conspecific skulls. For example, the characteristic formation of the frontal bone and the temporal arches of the non-conspecifics would be markedly different from those of the conspecifics. Therefore, the area of image covered by the stimuli would not be controlled for each species. This particular concern about the area of stimuli presented is somewhat reflected in Fig 2(c) skull condition where the rat-skull takes up more longitudinal area than the others. Fig 3 confirms this, where we see that looking duration on the rodent skull in lateral condition was indeed significantly higher than on the rest.

Now, to test hypothesis 1 of this study, the authors are failing their precondition: "With different sets of skulls, all else being the same...", because apart from image size, all other factors are not being the same. On the other hand, does one need to control for the skull-size at all, if the goal is to test whether chimpanzees recognize their conspecific skulls? The difference in skull sizes between non-specifics could perhaps be controlled by including skulls of infant or juvenile chimpanzees into the stimuli-set.

My suggestions would be, the authors should justify, their decision to correct for size of the images and not dimensions of the skulls.

Section 4.1.4 Teeth & Mouth Regions: The authors find that chimpanzees looked longer at the teeth region of conspecific skulls. This could be an outcome of exposed teeth in fear-grimace facial expressions of chimpanzees, therefore, could be a confounding result. We are not sure if the chimpanzees are paying attention to a particular emotion, i.e. fear indicated by exposed teeth, or the teeth area of a skull.

Section 5.2 Framing the results within face processing research: As pointed out earlier, this sections need reflection on previous eye-tracking studies on face recognition by chimpanzees (e.g. by Kano et al), how finding of the present study relate to previous results, and how this manuscript goes beyond previous findings, thereby establishing its novelty.

Reviewer: 4

Comments to the Author(s)

The manuscript presents an interesting finding. It advances the literature on how chimpanzees process face-related information by presenting apes with different species' skulls, faces, and skull-shaped stones. The study also complements previous literature on how chimpanzees react to dead conspecifics. However, although the study presents the results in a very detailed manner, I propose an alternative way to analyze the results (below). Furthermore, the study misses many interesting references related to natural encounters of chimpanzees with dead conspecifics. Finally, the study needs to clarify other points before acceptance. Below, I provide detailed comments. Unfortunately, the article is not number lined; thus, I can only refer to the main sections to target the comments.

Summary

Break down the very last sentence for a better flow. As of now, it contains six commas, and it is a bit difficult to follow.

Introduction

The authors start their introduction by discussing both chimpanzee and elephant literature. Then suddenly, after the sentence starting "Even more,.." they only refer to elephants and continue referring to non-human animals in general. I would restructure this first part of the introduction, perhaps moving from general findings across taxa to specific chimpanzee-elephant comparisons. At the beginning of page 3, the authors cite Halder & Schenkel 1972 as a "fairly obscure paper". Either elaborate on that point or remove that adjective.

Chimpanzees

I am a bit confused with this section of the introduction. The author cites two classic experiments but misses the opportunity to cite much more recent and perhaps, relevant work by authors

including van Leeuwen, Cronin or Biro to name a few. It seems that most of the literature on chimpanzees' natural encounters with dead bodies is missing. Authors should include crucial references, especially given the limited amount of literature in this field.

Cronin, K. A., Van Leeuwen, E. J., Mulenga, I. C., & Bodamer, M. D. (2011). Behavioral response of a chimpanzee mother toward her dead infant.

Van Leeuwen, E. J., Mulenga, I. C., Bodamer, M. D., & Cronin, K. A. (2016). Chimpanzees' responses to the dead body of a 9-year-old group member. *American Journal of Primatology*, 78(9), 914-922.

Subjects

The authors mention that there were three dropouts. Please state why this was the case.

Procedure and stimuli

The authors have counterbalanced very well all combinations of condition, sub-conditions, and species. However, I believe the paragraph needs clarification. At first sight, it is hard to find where the 144 trials come from. I would suggest the authors do something similar to what they are already doing in the Statistical Analysis section (e.g., information in brackets: 4 species X 3 conditions X 3 sub-conditions X 4 variations for a total of 144 trials).

Out of curiosity, I wonder if researchers considered including human faces to control for familiarity given how unfamiliar they were to all animals except conspecifics.

I found a bit unclear which conditions did they received on each testing day. For example, were face and skull conditions always presented together with stones? In other words, days with face and stone and days with skull and stone? I believe this is not the case, and every day they received either skulls or faces or stones, but I am not entirely sure.

Statistical Analysis

I was puzzled that the authors did not statistically analyze whether chimpanzees look significantly longer to faces vs. skulls vs. skull-shape stones. For instance, if chimpanzees' average looking time is not significantly longer for skulls than skull-shape stones, the significance of the overall results may change.

To do that, the authors could try LMM with a Gaussian error structure. A linear mixed model would allow authors to study in the same model the effects of species, condition (skull, face, skull-shaped stone) and sub-conditions together with any interaction between these variables. The authors could also control for any learning and order effects by including session and/or trial numbers. Such a model would also allow authors to not just control for subject ID as a random effect (as in simple ANOVAs) but also the random slopes between the random effect (subject) and the main effects (e.g. condition, species, sub-conditions). The package the authors use is suitable for the proposed analysis.

Notice, though, that I am not stating that the current analyses are incorrect. On the contrary, the authors could extract more information from their data while better controlling the effect of specific variables. Furthermore, the authors would reduce the amount of ANOVAs (eight if I am not mistaken) to just a couple of models focusing on overall looking times and on specific regions of interest within the chimpanzee sub-sample.

Results

Overall looking time

I am not entirely sure why the authors need the first paragraph of the result section. If this is a requisite of the journal, it is ok, but it feels like part of the discussion. It is anticipating the results that are later presenter in greater detail with statistical values.

This first section also needs some rephrasing in any case. The authors mention that there appears to be a significant trend across skull face and stone conditions regarding stimuli in frontal sub-conditions. This is right, but then I do not understand why right after this sentence, the authors reiterate that there was more substantial attention to diagonal and frontal (again) sub-conditions in both skull and face but diminished in the lateral sub-conditions. I think it is better to first present the effect on the frontal sub-condition at once.

The overall looking time section seems to introduce the results. I would clarify here why the authors only considered diagonal and frontal chimpanzee faces and skulls for investigating the stimuli chimpanzee look at. The authors only explain that by the end of the results section.

Skulls and faces

In these two results sections, the authors mention, by the end of both paragraphs, that "all other comparisons were non-significant". Do the authors refer to comparisons within the lateral sub-conditions or in general? The statement seems to be especially unclear for the face sections.

Teeth and mouth

I wondered what was the reason not to show animal faces (or at least chimpanzee faces for this analysis) with the mouth open so that teeth were visible? Was it due to the potential emotional reactions they elicit (e.g., fear grin face in chimpanzees)? I understand the authors cannot control for eye presence in skulls, but it is possible to show teeth in alive faces. It might be worth clarifying the reasons for future readers.

Discussion

General findings

The sentence starting with "Moreover, compared to..." is very unclear. I would recommend reformulating it. Similarly, the sentence discussing the non-significant difference between chimpanzees' and cats' frontal skulls is hard to follow.

The authors state that for the lateral skull sub-condition, there were no significant differences among stimuli. Please correct the statement. They seem to look longer for rats than all other species – and there are significant differences between rats and dogs.

The end of the second discussion paragraph needs references.

Further considerations

The end of this section is slightly confusing. It needs rephrasing (e.g., ..they previously seen dead..).

Conclusions

The sentence starting "For H1;.." needs rephrasing.

For H2, why is the non-significant trend between chimpanzees and rats' frontal faces likely to disappear with a larger sample size? Curious to know why this is the only trend interpreted that way.

Finally, by the end of the manuscript (page 9) the authors discuss that the hypothetical face-module develop within the context of frontal face-to-face interactions for the first time. Before, they only refer to this module in general terms. Could you elaborate why it evolved in the context of face-to-face interactions? As of now it reads as a conclusion stemming solely from your results.

Reviewer: 5

Comments to the Author(s)

The authors present a novel eye-tracking study in chimpanzees that examines visual attention to pictures of faces, skulls, and stones (that mimicked the shape/color of the species). Four different species of mammals were presented: chimpanzee, cat, dog, and rat, and each condition was presented at three different angles (diagonal, frontal and lateral). As such, this is an impressive dataset for examining heterospecific and conspecific face perception in chimpanzees along three main lines of inquiry: 1) looking times to conspecific versus heterospecific faces (which is not in itself novel, but the inclusion of these particular mammalian species is), 2) whether looking times change with angle of presentation and if so, whether the changes are consistent across con/heterospecifics or not, 3) whether progressive 'degrading' of the face signal (from face to skull to stone) alters looking times, and whether this is consistent across con/heterospecifics, 4) what are the salient features of the presented cues. Unfortunately, rather than using the theoretical framework of face processing here, the authors attempted to ground their research in comparative thanatology, which is not the most appropriate framing here. Consequently, I do not think this manuscript is publishable in its current form. If the authors were willing to

substantially revise the manuscript, including the theoretical framework, then I think it could be a publishable contribution.

The manuscript should also be edited for clarity/word choices throughout. There are some unnecessarily long and confusing sentences (e.g. the last sentence of 1. Summary).

Specific comments:

Introduction:

As described above, the thanatological framework is weak. The authors themselves note that it is extremely rare for chimpanzees to come across conspecific skulls, so I am a bit confused as to why they chose to justify this study in this manner. While this is worth some treatment in the discussion, there is a far more rich literature on how chimpanzees and other primates process faces that provide appropriate background and justification for the study including:

For comparing looking times to con- and heterospecific faces

- 1) Kano and Tomonaga 2009 - chimpanzee viewing patterns when shown pictures of chimpanzees, humans, other mammals
- 2) Hattori et al. 2010 - differential sensitivity to conspecific and allospecific cues in chimpanzee and humans

Angles of presentation/degradation of signal:

- 1) Tomonaga & Imura 2009
- 2) Dahl et al. 2013
- 3) Taubert et al. 2012

4) Herman et al. 1990 (in dolphins!)

Face scanning/salient features - many, but including

- 1) Hirata et al. 2010
- 2) Kano et al. 2015
- 3) Kano et al. 2012

Methods:

Please provide more description on calibration methods, I do not know what "two small clips on each referent point" means.

Figure 1. is quite confusing since the stimuli in front of the chimpanzee in the figure looks nothing like the examples provided in Figure 2, please clarify.

Figure 2. Please rework this to show the species in the same configuration across all conditions (e.g. chimpanzee top left, dog top right, cat bottom left, rat bottom right). I understand that this is not how they were presented to subjects (to avoid positional biases), but it would be useful to be able to match the stone condition directly to the species/angle of interest.

More information needs to be provided on the statistical methods. What does AOI (species-stimuli) refer to? Is this, for example, cat-face (vs cat-skull) or cat-frontal versus cat-lateral? It seems like it is the skull/face/stone category differences later in the paragraph, but then I don't understand how the angle of presentation was analyzed. Please clarify this so that your work is replicable.

Results:

Please provide descriptive statistics (mean looking times and sds) for all comparisons.

Please also refer to the appropriate figures in the text of the results.

A summary table of results would be much more digestible.

Discussion:

The discussion should be reworked according to the suggested framing above.

===PREPARING YOUR MANUSCRIPT===

===PREPARING YOUR REVISION IN SCHOLARONE===

Author's Response to Decision Letter for (RSOS-210349.R0)

See Appendix A.

RSOS-210349.R1 (Revision)

Review form: Reviewer 1

Is the manuscript scientifically sound in its present form?

Yes

Are the interpretations and conclusions justified by the results?

Yes

Is the language acceptable?

Yes

Do you have any ethical concerns with this paper?

No

Have you any concerns about statistical analyses in this paper?

No

Recommendation?

Accept as is

Comments to the Author(s)

Thank you for your careful attention to the reviewers' comments in the last round.

Review form: Reviewer 3

Is the manuscript scientifically sound in its present form?

Yes

Are the interpretations and conclusions justified by the results?

Yes

Is the language acceptable?

Yes

Do you have any ethical concerns with this paper?

No

Have you any concerns about statistical analyses in this paper?

No

Recommendation?

Accept as is

Comments to the Author(s)

The authors have satisfactorily addressed my questions and concerns with their previously submitted manuscript. I have no further comments on the present one.

Review form: Reviewer 4

Is the manuscript scientifically sound in its present form?

Yes

Are the interpretations and conclusions justified by the results?

Yes

Is the language acceptable?

Yes

Do you have any ethical concerns with this paper?

No

Have you any concerns about statistical analyses in this paper?

No

Recommendation?

Accept with minor revision (please list in comments)

Comments to the Author(s)

The authors have significantly improved an already interesting manuscript during the review process. However, I still have a few comments about the interpretation of the results that would need clarification before my final acceptance.

My main comment concerns the interpretation and analysis of the results. It is important to show that chimpanzees fixated significantly more on chimpanzee faces than on chimpanzee skulls. This would support the authors' interpretation stating that chimpanzees may interpret skulls as degraded faces. As of now, it is not easy to see that this has occurred. The interaction effect between type and species in the frontal and diagonal models supports that chimpanzees fixated more in conspecifics than in other species (with some exceptions) for frontal and diagonal faces and skulls, but not for stone-shaped images. But there seems to be no apparent difference in their fixation patterns for chimpanzee faces vs. skulls with other species.

One possibility to check faces & skulls & stones (at least in chimpanzees) is to include chimpanzee species and test the main effect of type. Then, using the lme4 package, the authors could use the drop1 function to assess pair-wise comparisons between type levels. Re-leveling the model variables (e.g., species) and using the drop1 function might be better to compare chimps against other species instead of using the Bonferroni correction.

Page 19 of 55 (the document I got from RSOS), L 16: In section 3.3, it is unclear that apes experienced only one type of stimuli (face, skull, or stone) within a single image group. The authors wrote, "each image group had three types.", however, it seems that each image group was composed of four pictures, one per species, and all of them were of the same type (either face, skull, or stone) – as shown in Figure 2.

Page 19 L 48: When you refer to three independent tests (one for each angle), do you mean three independent LMM models, correct?

Page 20 L 7: The presentation of the results is slightly inconsistent across sections. For instance, when the authors state that the "fixation durations were significantly longer overall for the chimpanzees," I would instead state that the fixation durations were significantly longer for chimpanzees compared to or versus other species. The authors already do something similar to what I propose in L 30.

Page 21 L 42: Either use versus or versus.

Page 21 L 48: The sentence needs to be rephrased "this suggests the stones, with only outlines of each species, were too degraded a stimuli..."

Page 21 L 55: The authors suggest that fixation times were slightly higher in diagonal faces compared to diagonal skulls. Do they refer to fixation times for all stimuli or just for chimpanzees? This is the kind of comparison that would support the faces & skulls & stones, at least for diagonal angles, and it would be interesting to know whether those differences are

significant. Table 3 seems to support this for species chimpanzee, although there is a high variation for chimpanzee diagonal faces fixation times.

Page 22 L 8: I would suggest the authors include the unpublished results in the Electronic Supplementary Material. This analysis supports the authors' interpretation of the lateral fixation times for rat skulls.

Page 23 L 10: The sentence needs some revision "one possible explanation is the result was due..."

Page 25 L 19: The sentence in brackets is not completely accurate. The frontal conditions had overall longer looking times than diagonal conditions, but for instance the fixation time for diagonal chimpanzee face is longer than for frontal chimpanzee face.

Review form: Reviewer 5

Is the manuscript scientifically sound in its present form?

Yes

Are the interpretations and conclusions justified by the results?

Yes

Is the language acceptable?

No

Do you have any ethical concerns with this paper?

No

Have you any concerns about statistical analyses in this paper?

No

Recommendation?

Reject

Comments to the Author(s)

This revision of the submitted paper is much improved in terms of clarity, especially with regards to the statistical approach and the results. However, I still think it is too heavily framed in the introduction as a thanatological study (which it is not) versus a face perception study. Aside from the abstract/summary, a reader of the introduction as written would assume that they were about to read about a study in which various skulls were placed in a testing area of the PRI and chimpanzees attention to and interaction with those skulls was measured. It is not until the second paragraph of section 2.3 that the reader becomes aware that this is an eye-tracking study, at which point, many readers would think, "Wait, there is a large literature on conspecific and cross-species face perception, including using eye-tracking, so how is this study situated in that literature?". My recommendation would be to completely reframe the introduction to focus on the face-processing and eye-tracking literature that is now not reviewed until the discussion. The link to thanatology is intriguing, but this study does not tell us anything about what chimpanzees know, feel or do in response to death. Given that the main results are 1) chimpanzees prefer to look at chimpanzee faces over dog, cat and rat and 2) that this preference transfers to a partially degraded face signal (skulls), but not to a more degraded signal (stones), and 3) teeth seem particularly salient, I think this paper is more appropriate for a field specific journal, such as Journal of Comparative Psychology or Animal Cognition.

Section and line specific comments are below:

1. Summary (Abstract)

This reads as if it is prefacing two studies – one about thanatology and one about face processing
 Lines 32-33 – Use “hypothesize” or “predict” rather than “suspect” and then state your predictions clearly, otherwise the sentence that begins “supporting our hypotheses” is unclear.

Line 33 – I would use “preferred” or “attended to” rather than “attracted”

Line 34-35 – In the sentences with numbered results add the direction of the effect “chimpanzees attended significantly MORE towards conspecific skulls”

Line 57-59 – I would remove the last sentence of this paragraph and just add the Halder reference to line 54 as so (Hutson et al., 2013, but see Halder & Schenkel, 1972).

Introduction, Section 2.2 (page 3)

Line 31 – Insight into what question?

Line 32 – “researched the comparative development of chimpanzees and humans” - you do not need to state an author’s field of study

Line 44-46 – This sentence leaves out an important category of chimpanzee reactions, namely maternal responses to corpses (Biro et al. 2010, Lonsdorf et al. 2020, Carter et al. 2021)

The introduction still does not integrate the face processing literature at all.

Section 3.3 Procedure

Page 5, Line 20 - I know that face-processing researchers are concerned with controlling for the size of stimuli. However, in this case, you should consider prioritizing ecological validity given that you are trying to gain insight into a process that happens in nature. That is, chimpanzee faces and skulls are naturally bigger than cat, mouse and (depending on the breed) dogs, so it’s possible that the chimpanzees were attentive to chimpanzees faces and skulls due to the novelty of those items being presented much smaller than the natural size, or at least the natural proportional size to a mouse face/skull.

Results/figures

In figures 2, 4, 5, 6, I recommend reordering the X-axis so that face, skull, stone, is the order going from left to right, which follows the path of degradation

Discussion, Section 5

Page 7

Line 19 – Missing word between “find” and “if”

Line 26-27 – If I am understanding your statistical results correctly, then you cannot say that there was any significant preference for conspecific skull-shaped stones.

Page 8

Line 19-22 – Yes, this is the main finding, that chimpanzee skulls are likely detected as degraded faces. I recommend rewording this as skulls do not present themselves.

Section 5.2 – much of this belongs in the Introduction, especially the first four paragraphs

Line 36-50 – This paragraph on neuroimaging findings seems outside the scope of this paper, I recommend excluding it.

Section 5.3 – while interesting, does not have much relevance for what was actually tested. The only portion that is relevant for this paper is the paragraph on tusks and teeth, but this could be much reduced and included in the face-processing portion.

Conclusion & Future Directions, Page 11

I found these two paragraphs much more relevant and explicitly linked to the experimental paradigm, and therefore much more compelling.

Decision letter (RSOS-210349.R1)

Dear Mr Goncalves

On behalf of the Editors, we are pleased to inform you that your Manuscript RSOS-210349.R1 "Staring death in the face: Chimpanzees' attention towards conspecific skulls and the implications of a face-module guiding this behavior" has been accepted for publication in Royal Society Open Science subject to minor revision in accordance with the referees' reports. Please find the referees' comments along with any feedback from the Editors below my signature.

Please submit your revised manuscript and required files (see below) no later than 7 days from today's (ie 08-Nov-2021) date. Note: the ScholarOne system will 'lock' if submission of the revision is attempted 7 or more days after the deadline. If you do not think you will be able to meet this deadline please contact the editorial office immediately.

on behalf of Essi Viding (Subject Editor)
openscience@royalsociety.org

Associate Editor Comments to Author

Associate Editor: 1

Comments to the Author:

Thank you for the substantial revisions, which have improved the manuscript. As you'll see, the views of several of the reviewers are in favour of publication; however, two of the reviewers offer a range of comments that you'll need to address before the Editors will be prepared to accept the work. In particular, reviewer 5 in the decision letter has a number of substantial concerns regarding the framing of your work in the wider context of existing literature - their concerns regarding the journal's suitability for the work are not criteria on which your work will be judged at this stage, but they make reasonable comments regarding how you are situating your work, which you must address. As two reviewers comment that the work would benefit from further language polishing, please do seek advice from a service such as those available at <https://royalsociety.org/journals/authors/benefits/language-editing/> before resubmitting. If you need a little longer than the usual 7-day turnaround, please let the editorial office know, and they will be able to assist.

Reviewer comments to Author:

Reviewer: 3

Comments to the Author(s)

The authors have satisfactorily addressed my questions and concerns with their previously submitted manuscript. I have no further comments on the present one.

Reviewer: 1

Comments to the Author(s)

Thank you for your careful attention to the reviewers' comments in the last round.

Reviewer: 4

Comments to the Author(s) (See also attachment 'Review RSOS.pdf')

The authors have significantly improved an already interesting manuscript during the review process. However, I still have a few comments about the interpretation of the results that would need clarification before my final acceptance.

My main comment concerns the interpretation and analysis of the results. It is important to show that chimpanzees fixated significantly more on chimpanzee faces than on chimpanzee skulls. This would support the authors' interpretation stating that chimpanzees may interpret skulls as degraded faces. As of now, it is not easy to see that this has occurred. The interaction effect between type and species in the frontal and diagonal models supports that chimpanzees fixated more in conspecifics than in other species (with some exceptions) for frontal and diagonal faces and skulls, but not for stone-shaped images. But there seems to be no apparent difference in their fixation patterns for chimpanzee faces vs. skulls with other species.

One possibility to check faces > skulls > stones (at least in chimpanzees) is to include chimpanzee species and test the main effect of type. Then, using the lme4 package, the authors could use the drop1 function to assess pair-wise comparisons between type levels. Re-leveling the model variables (e.g., species) and using the drop1 function might be better to compare chimps against other species instead of using the Bonferroni correction.

Page 19 of 55 (the document I got from RSOS), L 16: In section 3.3, it is unclear that apes experienced only one type of stimuli (face, skull, or stone) within a single image group. The authors wrote, "each image group had three types.", however, it seems that each image group was composed of four pictures, one per species, and all of them were of the same type (either face, skull, or stone) – as shown in Figure 2.

Page 19 L 48: When you refer to three independent tests (one for each angle), do you mean three independent LMM models, correct?

Page 20 L 7: The presentation of the results is slightly inconsistent across sections. For instance, when the authors state that the "fixation durations were significantly longer overall for the chimpanzees," I would instead state that the fixation durations were significantly longer for chimpanzees compared to or versus other species. The authors already do something similar to what I propose in L 30.

Page 21 L 42: Either use versus or versus.

Page 21 L 48: The sentence needs to be rephrased "this suggests the stones, with only outlines of each species, were too degraded a stimuli..."

Page 21 L 55: The authors suggest that fixation times were slightly higher in diagonal faces compared to diagonal skulls. Do they refer to fixation times for all stimuli or just for chimpanzees? This is the kind of comparison that would support the faces > skulls > stones, at least for diagonal angles, and it would be interesting to know whether those differences are significant. Table 3 seems to support this for species chimpanzee, although there is a high variation for chimpanzee diagonal faces fixation times.

Page 22 L 8: I would suggest the authors include the unpublished results in the Electronic Supplementary Material. This analysis supports the authors' interpretation of the lateral fixation times for rat skulls.

Page 23 L 10: The sentence needs some revision “one possible explanation is the result was due...”

Page 25 L 19: The sentence in brackets is not completely accurate. The frontal conditions had overall longer looking times than diagonal conditions, but for instance the fixation time for diagonal chimpanzee face is longer than for frontal chimpanzee face.

Reviewer: 5

Comments to the Author(s)

This revision of the submitted paper is much improved in terms of clarity, especially with regards to the statistical approach and the results. However, I still think it is too heavily framed in the introduction as a thanatological study (which it is not) versus a face perception study. Aside from the abstract/summary, a reader of the introduction as written would assume that they were about to read about a study in which various skulls were placed in a testing area of the PRI and chimpanzees attention to and interaction with those skulls was measured. It is not until the second paragraph of section 2.3 that the reader becomes aware that this is an eye-tracking study, at which point, many readers would think, “Wait, there is a large literature on conspecific and cross-species face perception, including using eye-tracking, so how is this study situated in that literature?”. My recommendation would be to completely reframe the introduction to focus on the face-processing and eye-tracking literature that is now not reviewed until the discussion. The link to thanatology is intriguing, but this study does not tell us anything about what chimpanzees know, feel or do in response to death. Given that the main results are 1) chimpanzees prefer to look at chimpanzee faces over dog, cat and rat and 2) that this preference transfers to a partially degraded face signal (skulls), but not to a more degraded signal (stones), and 3) teeth seem particularly salient, I think this paper is more appropriate for a field specific journal, such as *Journal of Comparative Psychology* or *Animal Cognition*.

Section and line specific comments are below:

1. Summary (Abstract)

This reads as if it is prefacing two studies – one about thanatology and one about face processing

Lines 32-33 – Use “hypothesize” or “predict” rather than “suspect” and then state your predictions clearly, otherwise the sentence that begins “supporting our hypotheses” is unclear.

Line 33 – I would use “preferred” or “attended to” rather than “attracted”

Line 34-35 – In the sentences with numbered results add the direction of the effect “chimpanzees attended significantly MORE towards conspecific skulls”

Line 57-59 – I would remove the last sentence of this paragraph and just add the Halder reference to line 54 as so (Hutson et al., 2013, but see Halder & Schenkel, 1972).

Introduction, Section 2.2 (page 3)

Line 31 – Insight into what question?

Line 32 – “researched the comparative development of chimpanzees and humans” - you do not need to state an author’s field of study

Line 44-46 – This sentence leaves out an important category of chimpanzee reactions, namely maternal responses to corpses (Biro et al. 2010, Lonsdorf et al. 2020, Carter et al. 2021)

The introduction still does not integrate the face processing literature at all.

Section 3.3 Procedure

Page 5, Line 20 - I know that face-processing researchers are concerned with controlling for the size of stimuli. However, in this case, you should consider prioritizing ecological validity given that you are trying to gain insight into a process that happens in nature. That is, chimpanzee faces and skulls are naturally bigger than cat, mouse and (depending on the breed) dogs, so it’s possible that the chimpanzees were attentive to chimpanzees faces and skulls due to the novelty of those items being presented much smaller than the natural size, or at least the natural proportional size to a mouse face/skull.

Results/figures

In figures 2, 4, 5, 6, I recommend reordering the X-axis so that face, skull, stone, is the order going from left to right, which follows the path of degradation

Discussion, Section 5

Page 7

Line 19 – Missing word between “find” and “if”

Line 26-27 – If I am understanding your statistical results correctly, then you cannot say that there was any significant preference for conspecific skull-shaped stones.

Page 8

Line 19-22 – Yes, this is the main finding, that chimpanzee skulls are likely detected as degraded faces. I recommend rewording this as skulls do not present themselves.

Section 5.2 – much of this belongs in the Introduction, especially the first four paragraphs

Line 36-50 – This paragraph on neuroimaging findings seems outside the scope of this paper, I recommend excluding it.

Section 5.3 – while interesting, does not have much relevance for what was actually tested. The only portion that is relevant for this paper is the paragraph on tusks and teeth, but this could be much reduced and included in the face-processing portion.

Conclusion & Future Directions, Page 11

I found these two paragraphs much more relevant and explicitly linked to the experimental paradigm, and therefore much more compelling.

===PREPARING YOUR MANUSCRIPT===

one version should clearly identify all the changes that have been made (for instance, in coloured highlight, in bold text, or tracked changes);

===PREPARING YOUR REVISION IN SCHOLARONE===

-- If you are requesting an article processing charge waiver, you must select the relevant waiver option (if requesting a discretionary waiver, the form should have been uploaded, see 'File upload' above).

-- If you have uploaded any electronic supplementary (ESM) files, please ensure you follow the guidance at <https://royalsociety.org/journals/authors/author-guidelines/#supplementary-material> to include a suitable title and informative caption. An example of appropriate titling and captioning may be found at https://figshare.com/articles/Table_S2_from_Is_there_a_trade-off_between_peak_performance_and_performance_breadth_across_temperatures_for_aerobic_scope_in_teleost_fishes_/3843624.

Author's Response to Decision Letter for (RSOS-210349.R1)

See Appendix B.

Decision letter (RSOS-210349.R2)

Dear Mr Goncalves,

It is a pleasure to accept your manuscript entitled "Staring death in the face: Chimpanzees' attention towards conspecific skulls and the implications of a face-module guiding this behavior" in its current form for publication in Royal Society Open Science. The comments of the reviewer(s) who reviewed your manuscript are included at the foot of this letter.

Please see the Royal Society Publishing guidance on how you may share your accepted author manuscript at <https://royalsociety.org/journals/ethics-policies/media-embargo/>. After publication, some additional ways to effectively promote your article can also be found here

<https://royalsociety.org/blog/2020/07/promoting-your-latest-paper-and-tracking-your-results/>.

on behalf of Prof Essi Viding (Subject Editor)
openscience@royalsociety.org

Appendix A

General Reply:

The authors would like to thank the five reviewers for their helpful comments. We strove to accommodate each of points made and paused for reflection when these involved changing substantially the manuscript. We believe the final product is much improved from the first version. We would also like to extend our thanks to the Editorial Coordinators Drs Lianne Parkhouse and Anita Kristiansen for their kind assistance throughout.

Reviewer: 1

Comments to the Author(s)

This is a really important, fascinating and understudied topic, which the authors should be commended for pursuing. The stimuli categories are nicely controlled. I have only minor clarification comments.

Q1: Why were the particular species of skulls chosen? How were the stones created/selected?

R1: We chose those three species (dogs, cats and rats) for two reasons: the first reason was practicality (there were many pictures available of skulls of these species permitting the amount of sub-conditions), the second was they were all fairly distinct from each other and distinct from chimpanzees' skulls. The stones were manipulated to fit the shape of each species' skull, we make this point now in the methods.

Q2: Within the main text, the presentation of stimuli needs to be more detailed. How were the stimuli presented – in pairs, one at a time? How many per session? How balanced per order? What did the chimpanzees do on each trial? How many sessions were presented to each subject?

R2: We included a new figure detailing the presentation of stimuli. We also made it clearer in the text.

Q3: Why did three of the subjects drop out?

R3: Since the explanation is a bit long, we included the information as a footnote in the manuscript: *“These dropouts were due to different reasons (neurological, physical and temperamental) that affected the completion of the sessions. One (Pendesa) exhibited erratic visual patterns, she suffers from an arachnoid cyst in her brain which seems to have partially affected her visual field (see Kaneko et al., 2013). Another (Mari), has difficulty breathing through her nose which affected the session as she would choke on the juice while trying to breathe through her mouth, at times resulting in her abandoning the experiment altogether. The last one (Pan), would for no apparent reason abandon the experiment midway or not come altogether. Different approaches were attempted to coax her into the eye tracking session without satisfactory results.”*

Q4: The structure for the analysis is unclear. Why was AOI one factor – why not species and material (skull, stone)? What are the interactions in the model?

R4: We have extensively rewritten this in the methods section to include the test of those interactions.

Q5: The results are similarly not written up in a clear manner. The authors need to refer explicitly to significant main effects and interactions and specify the factors more clearly. How can there be a significant effect between the DV and AOIS?? (line 53)

R5: The line in question has been corrected and that section rewritten.

Q6: Why analyze the faces, skulls and stones separately? Was there an interaction that called for splitting the analysis this way?

R6: We reanalyzed them together now by angle.

Q7: It would have been interesting to see whether there was the same conspecific preference for complete skeletons except for skulls, which would negate the ‘face attention’ explanation.

R7: This is a noteworthy point. Besides the point of asking if there is a faceness aspect to the skull, the reviewer implicitly suggests, is there a bodyness to the skeleton? Judging from the biological motion and body posture research it’s likely they do. But would such an experiment precisely negate any “face attention” and not act more as an addition to it (Assuming the chimps show such interest)? Perhaps having conditions of skeletons with and without skulls? Another thing to consider is the ecological aspects: 1) while chimpanzee skulls are usually found almost intact in the forest, cases where a full skeleton is complete (as described by Watts) are extremely rare because of scavengers; 2) using images of skeletons for this condition is tricky since skeletons are photographed in a life-like standing positions, actual chimpanzee skeletons in the forest are not, this could be problematic?

Q8: Why the preference for rat skulls in the lateral condition? I may have missed it but I didn’t think the authors addressed this in the discussion.

R8: In the new analysis with the lateral condition pooled this preference is no longer significant. We believed chimpanzees looked longer at the lateral rat skull because unlike the other skulls, it remains as if open (i.e. as if vocalizing or biting). When we ran a saliency analysis it showed this skull was slightly more salient than all others, even if we resized it to a point when it became the smallest of the 4 skulls. Moreover, when we covered the original rat skulls (painted bone and teeth over the open mouth area) our chimpanzees no longer looked at it significantly.

Q9: P. 9, line 8, I don’t follow the reference to second order relational info.

R9: We have extensively rewritten this section in the manuscript, second order relations are now clarified.

Q10: Line 32 “where” should be “were.”

R10: We are not sure where this wording is in P9 Line32. We checked all the 44 mentions “were” and “where” and corrected them.

Reviewer: 2

Comments to the Author(s)

This manuscript describes an interesting study that contributes to the small literature on how chimpanzees respond to faces and the smaller literature on how they respond to skeletal elements, especially skulls. I think the authors could substantially improve the work, though, and I will offer some criticisms and suggestion that I think can help them to do this.

Q1: First, I should say that I have great sympathy for colleagues who are not native English speakers, but who are required to write in English for most academic journals in their fields. I could not write publishable quality work in Portuguese or Japanese (or in any language other than English). However, the manuscript could be substantially shorter and many points would be clearer if the authors have someone who is a native English-speaker help them with copy editing. Also, eliminating some of the extra words and the repetition would make the work more concise and easier for readers to understand. For example, the last sentence of the Abstract contains five phrases set off by commas that precede a final statement. Much of the wording is redundant and the authors could omit it. Having a single, short sentence that makes a clear statement would be much better. This same logic applies to many other places in the manuscript.

R1: We thank the reviewer for the specific corrections to the manuscript. We also carefully followed the suggestions for improving the english and making the overall manuscript clearer.

Q2: Also, the authors should use past tense in reporting their results and then in describing them in the Discussion. This is partly they are reporting on events that occurred in the past. Importantly, also is because when we make statements like “chimpanzees do X”, instead of explicitly saying that “in this study, the chimpanzees did X”, we seem to be saying that the results apply to all chimpanzees, everywhere and at all times. I have made some copy-editing suggestions below.

R2: We changed the verbs in the discussion accordingly.

Q3: More substantively, I would like to see the authors give more attention to the literature on face recognition. The paper is not just about responses to skulls. The authors state in the Discussion (P. 7, line 59) that one of their goals was to assess whether attention to skulls and head-like shapes was “guided by underlying interest in conspecific faces”. Some of the work on “interest in faces” has involved investigating how well primates (including humans) can discriminate among individual faces, which was not a question the authors were trying to address. However, I think they would benefit by looking at Taubert’s work (Perception, 38: 343, 2009) on “holistic” processing by humans of non-conspecific faces (including those of chimpanzees. She found evidence that humans process chimpanzee faces holistically – as we do faces of conspecifics – but we seem not to do so for spider monkeys or gorillas. I think this has some relevance to the question of whether chimpanzees process skulls, face-shaped (or really head-shaped) objects, or heterospecific faces like they do chimpanzee faces. Likewise, Taubert et al. (2017, Animal Cognition 20:321) found evidence that chimpanzees, like humans, are better attuned to averaged facial images than to specific ones; I suspect that this also has some implications for the authors’ work. In a series of experiments that also used eye-tracking, Kano et al. (2018, PLoS One) found that chimpanzees presented with video images looked longer at the mouths than the eyes of conspecifics and of other primate species (so did humans), the opposite of the result from the present study. I realize that the stimuli were different, which might explain the different results. However, I think the authors need to use this paper in contextualizing their study and their results. They could also benefit by noting that Kano et al. found considerable inter-individual variation (although less for chimpanzees than for the other species in their study), some of it related to variation in rearing histories, and by echoing Kano et al.’s emphasis on the fact that many primates attend selectively to conspecific eyes and faces, follow gaze, etc. In fact,

we have a big literature on the importance of faces in primate social worlds. (Incidentally, Kano et al. found that bonobos looked longer at eyes than faces; they suggested that this difference from chimpanzees might have occurred because bonobos are more tolerant of eye contact.) I suggest also that the authors look at the methods Kano et al. used, which differed from their own; actually, the general research question was different, because Kano et al. were interested in differences among chimpanzees, bonobos, rhesus, orangutans, and humans, but I think that paper offers valuable information for comparison. Kano and Tomanaga (2010; Animal Behavior 79:227) also deserves attention.

R3: We reworked section 5.2 extensively to include relevant comparative literature on faces.

Q4: My sense is that the lack of attention to studies like these was because the authors chose to limit their comparisons mostly to the studies of elephants that they cite. Those comparisons are interesting, but I don't think they tell us much that is new about chimpanzees. In turn, this is partly because the authors list "evolutionary thanatology" as one of the manuscript's keywords, but they have not really focused on what their results add to the literature on evolution thanatology other than to note that their results showed that their subjects gave more attention to chimpanzee skulls than to those of other species and that their particular attention to the teeth area is consistent with the idea that this skull region most resembles the faces of living chimpanzees. I would like the authors to say more about what this means for evolutionary thanatology, especially with reference to Goncalves and Carvalho (2019), which I think is one the best contributions to this literature. Doing that in the Discussion instead of repeating results would make the paper more valuable.

R4: We reworked section 5.3 to include this point.

Q5: I also think the paper would benefit from a table that summarizes the hypotheses and predictions and then notes which predictions were supported. This would make it easier for readers to follow the presentations of all the statistical results.

R5: We added a few more tables in the results section to aid the reading.

Q6: Finally, I would like to see some discussion of the average overall times that the chimpanzees looked at faces, skulls, or stones. I realize that they did not design the study to investigate whether the chimpanzees looked longer at chimpanzees faces than at chimpanzee skulls or stones shaped like chimpanzee faces/heads. We might expect them to look longest at faces, although the face-like aspects of skulls combined with their novelty might have undermined such an effect. Perhaps they could use their data to investigate this issue, although trying to do a statistical analysis may not be valid. In any case, I noticed the following: mean looking time was shorter for the stone than the face or skull in all conditions; it was longer for the face than the skull in the diagonal condition, but the absolute difference was quite small, and it was approximately equal for the face and skull in the frontal condition. I think this deserves some comment.

R6: We extensively reworked this section now to analyze these together now.

Q6: Also, the subjects looked longer at chimpanzee faces than rat faces in only one of three conditions. Was this perhaps because of their experience seeing live rats?

R6: Chimpanzees looked longer at chimpanzee faces in all orientations. Does the reviewer perhaps mean the skulls? Chimpanzees looked longer conspecific skulls in diagonal (marginally more than cat skulls) and frontal, but looked more at the lateral rat skull. We suspect this is due to the morphology of the rat skull that makes it salient by appearing to be open (as if biting or vocalizing).

Q7: One general comment on methods and reporting of results: I have not yet been convinced that we should abandon p-values. However, if we use conventional statistics and report these values, we should

not report “trends”. Trends are non-significant, by definition, and we should report the results as “not significant”.

R7: We corrected these statements throughout the manuscript.

Specific comments (page numbers refer to pages in upper right of manuscript, e.g. 2 of 12):

Q1: P. 3, first full sentence: Omit “a fairly obscure paper” and just say “Halder & Schenkel (1972) found that, like elephants, bovids [which species?] responded more to bones of conspecifics than to those of other species.”

R1: We changed it to: *“However, one paper reported the behavioral responses in bovids (*Bos javanicus*) to conspecific and non-conspecific bones”.*

Q2: P. 3, line 5: Saying “without any benefit” isn’t necessary. How would elephants benefit by interacting with bones of conspecifics?

R2: We removed “without any apparent benefit” from the sentence. It is unclear why they do this. Most interactions of the sort similar to behaviors of affiliative nature, it could be that touching bones is a coping response/has a calming effect but there is no evidence for it at present.

Q3: P. 3, last sentence of 2.1: First, say that these authors presented bones to the elephants (and indicate that these were elephants in the wild), not that they presented elephants to the bones. Also, this sentence could be shorter and clearer. “Whereupon” should be “on which”. Make a second sentence that is simply “The elephants showed most interest in elephant bones”, then a final sentence “Rasmussen (ref.) obtained similar results for captive elephants.

R3: We made all the suggestion corrections, sentence now reads: *“conducted by Goldenberg and Wittemeyer (2020) on which they presented elephant, giraffe, and Cape buffalo bones to wild elephants. Those elephants showed most interest in elephant bones. Similar results were obtained in captive elephants (Rasmussen in Goldenberg & Wittemeyer, 2020).”*

Q4: P. 3, 2.2: Omit the first sentence and just say “Two classic studies provided insights into chimpanzee responses to skeletons.”

R4: We omitted the first sentence.

Q5: P. 3, line 51: This should be “in chimpanzees and gorillas”; “among” makes it sound like gorillas and chimpanzees were interacting with each other.

R5: We changed the sentence accordingly.

Q6: 2.3: This paragraph would benefit from some editing to make it shorter and clearer (especially lines 11-13).

R6: We removed extraneous sentences and words in these lines.

Q7: P. 5, line 6, “together with stone”: Your description of “image groups” and the figure depicting examples of these indicates that a group consisted of either four images of skulls, 4 images that each showed a different species, or four images of skull-shaped stones. “Together with stone” makes it sound like you showed images of stones at the same time as you showed images of faces or skulls. Please make it clear that you showed the stone image groups on every day of testing, but alternated face groups and skull groups.

R7: We included a figure detailing the presentation sequence of each stimuli per trial. We also made it clearer in the text-

Q8: P. 5, line 8: Omit “significant”.

R8: We omitted this part.

Q9: P. 5, line 15: Missing “were” after “AOIs”.

R9: We added “were” to the sentence, thank you.

Q10: P. 5, 37-39: Omit the material about “trends”. Alternatively, the most you can say is that the mean values differed, but not significantly.

R10: We reworked these throughout the manuscript.

Q11: P. 5, line 42: Make it clear what was “diminished”.

R11: We changed the sentence to: *“Longer looking durations were present in the skull and face conditions but diminished in the stone condition.”*

Q12: P. 5, line 42, sentence, starting “Since”: Omit this sentence; you already said this in the Intro.

R12: We omitted this sentence.

Q13: P. 5, line 43, sentence starting “For analyses”: this should be in Methods.

R13: We reworked this information now in the methods (statistical analyses).

Q14: P. 5, line 45: “are” should be “were”.

R14: We replaced the words accordingly.

Q15: P. 5, line 46: “who” should be “which”.

R15: We replaced the words accordingly.

Q16: P. 5, line 53: Either say “there was either say there was “a significant relationship” between these variables or that “AOI significantly affected gaze duration”.

R16: Sentence now reads: *“there was a significant relationship between gaze duration and AOIs”.*

Q17: P. 5, line 54: Here and in other places where you present results of statistical tests (especially Tukey results), it would better to write “Overall fixation durations were longer [better word than “higher”] for the chimpanzee skull than for all others (cat: $t = -4.18$, $p = 0.003$, Cohen’s $d=0.83$; dog: $t = 4.13$, $p<0.001$, Cohen’s $d=0.84$ ”, etc. You already stated that you used Tukey HSD tests; you don’t need to repeat that for each comparison. Also, you can “t” instead of “t-ratio” and don’t need to repeat “chimp/rat”, etc. when you start by indicating that you are presenting the results of comparisons between chimps (o chimp-like stones) and each of the other stimuli.

R17: We’ve made these changes.

Q18: P. 6, line 5: “positively non-significant” should be “positive, but non-significant”.

R18: We removed the extraneous word.

Q19: 4.1.2 and later presentation of statistical results: as I urged above, please re-write this to say that you found (or did not find) significant relationships between variables, then state the results of the Tukey HSD tests as I suggested for p. 5, line 54.

Q20: P. 6, line 19: Omit “once again”.

R20: We removed the extraneous wording.

Q21: P. 6, line 20: Please do not invoke “marginal” effects. The relationship was non-significant.

R21: We reworked this section completely.

Q22: P. 6, line 38, sentence starting “Because”: This should be in Methods. Also, you don’t need to repeat that you ran an ANOVA for the skull regions (line 40); just give the result.

R22: We moved this sentence to the end of 3.4 Statistical Analysis in Methods.

Q23: Discussion:

The first paragraph would benefit from shortening and some re-organization. This also applies to several of the later sections. The entire Discussion could be considerably shorter and more concise. For one thing, you don’t need to repeat so many of the results (a table would help; see my comment above).

R23: We removed the extraneous sentences from the first paragraph.

Q24: P. 6, line 60, “visible face cues”: how could they orient to non-visible cues? I think you mean “cues that are visible on faces”.

R24: We changed this sentence accordingly, it now reads: *“if that attention was concentrated on cues naturally visible on faces such as teeth”*

Q25: P. 7, lines 1-3: This needs some re-writing. Again, don’t invoke “trends”. Also, the stones were not “conspecifics”; you mean stones that resembled the heads of conspecifics (or heterospecifics).

R25: We reworked this sentence and others and put conspecific in quotations for the stone stimuli.

Q26: P. 7, line 4: re-write as “from skulls, which retain multiple face like features, but not objects that resemble only face outlines”.

R26: We changed the sentence to: *“chimpanzees can extract familiar face-like features from skulls which retain multiple face-like features but less so with objects (stones) resembling only face/skull outlines”*.

Q27: P7, line 6: “is” should be “was”, and I suggest you re-write this as “The effects were weaker for lateral orientations than for frontal and diagonal orientations for all stimuli, presumably because lateral orientations carry less information about faces”. (Skulls & stones do not carry information about facial expressions!)

R27: We changed the sentence and wording accordingly.

Q28: P 7, line 12, “higher looking times”: Better as “Looking times were longer for the cat skull...”

R28: Average looking times were longer for the chimp skulls compared to the cat. We conveyed this in the text while keeping the suggestion *“In the diagonal skull sub-condition, while they exhibited the longest looking durations toward chimpanzee skulls this was only significantly different when compared to dog skulls. Looking times were also long for the cat skull”*.

Q29: P. 7, line 25, sentence starting “Finally”: “even more decreased” should be “even shorter” and “among” should be “for”.

R29: Sentenced changed accordingly.

Q30: P. 7, next sentence: “this higher looking tendency among” should be “the longer looking time for”.
R30: Sentence changed accordingly. Here and elsewhere in the manuscript “higher looking” was replaced with “longer looking”, thank you.

Q31: P. 7, paragraph starting 34: I can understand your point, but please see if you can make it more clearly.

R31: We reworked the paragraph, broken up, removed and added a few sentences for clarity. It now reads: *“Regarding the specific areas of the chimpanzee skulls, chimpanzees looked longer at the teeth, followed by the nose area, then lastly by the eye regions. The looking patterns were significantly longer for teeth versus eye sockets, but not for teeth versus nose cavity. Nose cavities received the second most attention, possibly due greater similarity with conspecific noses. Chimpanzee noses not being protruded, display greater similarities to nose cavities, which might account for the latter having the second longest looking duration of the three areas. Eye sockets on the other hand, without the eyes, are the least similar to the chimpanzee eye regions, thus recruited the least amount of attention. These results support both McComb’s study where elephants exhibited a higher interest in elephant tusks and Watts’ prediction that chimpanzee skulls may represent iconic features of chimpanzee faces, notably by having teeth.”*

Q32: Section 5.2: The material on a proposed “face module” could use some up-dating.

R32: We updated this section extensively.

Q33: Section 5.3: Not all chimpanzees live in rainforests (line 19).

R33: This is a good point. We corrected it in the manuscript.

Q34: P. 8, lines 25-27: Skeletal remains are only collected at research sites, and not at all of them. Also, researchers do not find the carcasses of most deceased chimpanzees, and I suspect that chimpanzees are much more likely to encounter them than researchers are. Also, chimpanzees in the wild have experience handling both the faces (heads) and skulls of species on which they prey; monkeys comprise most prey, and their faces and skulls certainly resemble those of chimpanzees.

R34: Another very good point. We added this caveat to the main text in the discussion and along with a references where a chimp interact with monkey skulls: *“Chimpanzees are likely aware of conspecific skeletons in their environments and these encounters are severely underreported. This knowledge also extends to situations where mothers carry dead infants to the point of mummification/skeletonization (Biro et al., 2010). Species with similar anatomy such as monkeys are also potentially informative; Boesch and Boesch (1990) report on a juvenile chimpanzee with a colobus monkey skull using a tool to scoop out the brain, another case involved an adult female using sticks to clean the orbits of colobus skull after she finished eating the eyes. Moreover, in some parts of Africa, lowland gorillas are sympatric with chimpanzees, and gorilla skeletons exceedingly similar to chimpanzees, are likely encountered.”*

Q35: P. 8, line 34: Teeth are also visible when chimpanzees yawn (very common!), screams, or fight, and they are often visible while individuals are feeding.

R35: We added this information in the main text. It now reads: *“Chimpanzee skulls bear a resemblance to particular facial expressions such as the fear grin and pout face. Moreover, chimpanzees’ teeth become visible during screams, and also during yawning or feeding.”*

Q36: Section 6: You don’t need to repeat the results. Just give a summary of what you think your results mean.

R36: We removed sentences that overly repeated the results and gave more emphasis to the interpretation of the results.

Reviewer: 3

Comments to the Author(s)

The manuscript entitled “Staring death in the face: Chimpanzees’ attention towards conspecific skulls and the implications of a face-module guiding this behaviour” presents results from an investigation of how captive chimpanzees attend to images of conspecific skulls, compared to those of non-conspecifics. Having established the state of the art of how elephants respond to conspecific skulls, as well as pointing out a lack of comparable studies on chimpanzees, despite these two species sharing several socio-cognitive traits, the authors justify the rationale of their study as “we aimed to find out, if chimpanzees exhibited the similar interest patterns as observed with African elephants.” As their findings, the authors report that chimpanzees pay most attention to images of conspecific faces, followed by those of their skulls and skull-shaped stones. Moreover, chimps scan the conspecific faces and skulls from teeth-region upwards till the nose.

I have two major concerns and a minor one, regarding the theoretical premise of this study, as well as interpretation of its results and their implication.

Q1: a) In my opinion, it would be better if the authors invoke a theoretical rationale with respect to primate ethology and evolution to justify the choice of the subject of their study. Right now, the rationale sits heavily on elephant-studies. While it is a pertinent rationale on grounds of comparative cognitive research, I believe the manuscript could benefit from bringing in a theoretical framework explaining the adaptive value of an ability to recognise conspecific skulls in their natural environment, for chimpanzees in particular, and primates in general.

R1: We reworked the manuscript **substantially** and answer these points in sections 5.2. (face research) and 5.3. (ecological considerations).

Q2: b) Do the findings of this particular study add anything novel to our existing knowledge on face-perception by chimpanzees? I do not make this remark after having seen the results, on the contrary, the study design itself made me wonder what different results could be expected than from eye-tracking studies on conspecific face recognition by chimpanzees. Are the skull-stimuli, used in this study was a proxy of the face-stimuli. Are the hypotheses a roundabout way to ask “Do chimpanzees look longer at conspecific-faces than non-conspecific ones?” the latter had already been addressed previously, therefore, the authors need to establish the novelty of this study, especially in comparison to extant facial recognition studies on chimpanzees.

I think the seed of potentially addressing this concern is already sown by the authors in this manuscript where they reflect, “Not only was this general interest in skulls greater for the conspecific skulls, but the same trend was shown slightly higher for conspecific faces but lesser for conspecific stones, which suggests the main factor for their attraction are faces and particularly conspecific face-like stimuli. What this suggests is that chimpanzees are able to extract familiar face-like features from distinct non-face stimuli such as skulls which still retain said features and fading in conspecific stones where only the outline of a face is shown.” This point should be the main pitch of this manuscript, addressing a crucial question, “when does the face-ness of conspecific faces cease to exist for chimpanzees?” In focussing on this particular question, the manuscript potentially could address a very interesting problem in cognitive science, that of perception of essence of objects. The capacity to identify an object, to a limit, even when they have lost some of their component parts, also form building blocks of formation of concepts and has profound implications in larger fields of such as that of cognitive linguistics.

R2: We appreciate the reviewer in-depth comment and other such as these since they made us reflect on how to improve the manuscript. The main point is that faces represent the strongest signal and this signal is degraded in lateral angles, moreover skulls (a face-like object) are more degraded still and these signals are even more degraded with the angles culminating in stones (facial outline) with the lowest signal. We've rewritten the manuscript to make these connections more clear (methods section, discussion).

Q3: c) The third concern is an extension of the above one. It is important that the authors discuss their rationale and findings in the context of previous face-recognition studies, using eye-tracking technology, on chimpanzees. In section 5.2 they discuss about brain areas responsible for facial recognition in primates. However, what remains missing is a more relevant discussion relating to the studies by Fumihiro Kano and colleagues on eye tracking research on chimpanzee face recognition (more so as those studies were largely done in the same facility with common chimps as participants).

R3: We have included his and many more comparative facial recognition studies and reworked section 5.2.

Below are my specific comments for respective sections. Since the manuscript did not have line numbers specified, I used the subheadings of the sections, thereafter quoting the particular lines to add my comments on them.

Q1: *Section 2.3 Research motivation & questions: The authors mention, "Moreover, if such interest was guided by some sort of recognition, would chimpanzees exhibit similar looking times towards conspecific faces (hypothesis 2)?" : clarification necessary for the rather vague phrase, 'guided by some sort of recognition', especially while stating a hypothesis to be tested.*

R1: We reworked this sentence to make this point clearer.

Q2: Section 3.3 Procedure & Stimuli: I understand that the skull-sizes were controlled for, something that the authors rectified after McComb et al 2006. However, my concern lies with dimensions of the non-conspecific skulls. For example, the characteristic formation of the frontal bone and the temporal arches of the non-conspecifics would be markedly different from those of the conspecifics. Therefore, the area of image covered by the stimuli would not be controlled for each species. This particular concern about the area of stimuli presented is somewhat reflected in Fig 2(c) skull condition where the rat-skull takes up more longitudinal area than the others. Fig 3 confirms this, where we see that looking duration on the rodent skull in lateral condition was indeed significantly higher than on the rest.

Now, to test hypothesis 1 of this study, the authors are failing their precondition: "With different sets of skulls, all else being the same..." , because apart from image size, all other factors are not being the same. On the other hand, does one need to control for the skull-size at all, if the goal is to test whether chimpanzees recognize their conspecific skulls? The difference in skull sizes between non-specifics could perhaps be controlled by including skulls of infant or juvenile chimpanzees into the stimuli-set. My suggestions would be, the authors should justify, their decision to correct for size of the images and not dimensions of the skulls.

R2: We disagree. We suspected this longer looking effect was due to particularities of the rat skull (it looks open even when closed as if vocalizing or biting) rather than dimensions. When we ran a saliency analysis the rat skull was slightly more salient, when we decreased the skull size to become the smallest of all 4 skulls the effect was still present. Moreover, we reran the experiments with a covered mouth (photoshopped bone and teeth in the mouth opening), the chimpanzees no longer looked at it significantly. Since the new statistical analysis suggested by other reviewers ended up not making this effect visible (it became non-significant) we did not include these further analyses. We do still mention this result in the first section of the discussion.

Q3: Section 4.1.4 Teeth & Mouth Regions: The authors find that chimpanzees looked longer at the teeth region of conspecific skulls. This could be an outcome of exposed teeth in fear-grimace facial expressions of chimpanzees, therefore, could be a confounding result. We are not sure if the chimpanzees are paying attention to a particular emotion, i.e. fear indicated by exposed teeth, or the teeth area of a skull.

R3: We make this point in a later in the paper (section 5.3). Another reviewer mentioned teeth are exposed in other contexts (feeding, fighting, yawning). While it may be true chimpanzees are looking more at skulls due to a particular emotion (i.e. fear-grimace), it is a facial expression and does not necessarily contradict our prediction of a face-module guiding the captive chimpanzees' attention. Functionally speaking both explanations use "chimpanzee-like traits" as the center of the argument; either the teeth resemble those on a chimp face or the teeth resemble a specific chimpanzee facial expression, we extend this argument in the manuscript.

Q4: Section 5.2 Framing the results within face processing research: As pointed out earlier, this sections need reflection on previous eye-tracking studies on face recognition by chimpanzees (e.g. by Kano et al), how finding of the present study relate to previous results, and how this manuscript goes beyond previous findings, thereby establishing its novelty.

R4: We reworked this section and included critical papers from this literature. Our main point of this study ties together research in pareidolia, conspecific face bias, and comparative thanatology more clearly. This has been part of an ongoing agenda started with previous papers (Gonçalves & Biro, 2018; Gonçalves & Carvalho, 2019) joining formerly unrelated research in comparative cognition (biological motion, uncanny valley, theory of mind, thanatosis, high order reasoning, necrophobia) into the field of comparative thanatology to give it a stronger theoretical and empirical support (since it has generally relied on single case reports).

Reviewer: 4

Comments to the Author(s)

The manuscript presents an interesting finding. It advances the literature on how chimpanzees process face-related information by presenting apes with different species' skulls, faces, and skull-shaped stones. The study also complements previous literature on how chimpanzees react to dead conspecifics. However, although the study presents the results in a very detailed manner, I propose an alternative way to analyze the results (below). Furthermore, the study misses many interesting references related to natural encounters of chimpanzees with dead conspecifics. Finally, the study needs to clarify other points before acceptance. Below, I provide detailed comments. Unfortunately, the article is not number lined; thus, I can only refer to the main sections to target the comments.

Q1: Summary

Break down the very last sentence for a better flow. As of now, it contains six commas, and it is a bit difficult to follow.

R1: We turned the last sentence into two for added clarity.

Q2: Introduction

The authors start their introduction by discussing both chimpanzee and elephant literature. Then suddenly, after the sentence starting "Even more,.." they only refer to elephants and continue referring to non-human animals in general. I would restructure this first part of the introduction, perhaps moving from general findings across taxa to specific chimpanzee-elephant comparisons.

At the beginning of page 3, the authors cite Halder & Schenkel 1972 as a "fairly obscure paper". Either elaborate on that point or remove that adjective.

R2: While we understand the rationale for moving from general taxa to chimpanzee-elephant comparisons, it doesn't flow as well. This is because we start from a general comparative thanatology approach to a specific "bone-centered" responses to skeletons. That being said we moved the "Even more..." sentence describing the elephant species to the elephant section where it makes more sense. We also removed the extraneous wording in the Halder & Schenkel sentence.

Q3: Chimpanzees

I am a bit confused with this section of the introduction. The author cites two classic experiments but misses the opportunity to cite much more recent and perhaps, relevant work by authors including van Leeuwen, Cronin or Biro to name a few. It seems that most of the literature on chimpanzees' natural encounters with dead bodies is missing. Authors should include crucial references, especially given the limited amount of literature in this field.

Cronin, K. A., Van Leeuwen, E. J., Mulenga, I. C., & Bodamer, M. D. (2011). Behavioral response of a chimpanzee mother toward her dead infant.

Van Leeuwen, E. J., Mulenga, I. C., Bodamer, M. D., & Cronin, K. A. (2016). Chimpanzees' responses to the dead body of a 9-year-old group member. *American Journal of Primatology*, 78(9), 914-922.

R3: We included the references and more. Section now reads: *"Several published reports describe chimpanzee's reactions towards their dead which range from affiliative to aggressive and from quiet and passive to loud and expressive (Teleki, 1973; Hosaka et al., 2000; Anderson et al., 2010; Cronin et al., 2011; Stewart et al., 2012; Boesch, 2012; van Leeuwen et al., 2016; Pruett et al., 2017)."*

Q4: Subjects

The authors mention that there were three dropouts. Please state why this was the case.

R4: This point was made elsewhere, we added a footnote stating the reasons for each dropout.

Q5: Procedure and stimuli

The authors have counterbalanced very well all combinations of condition, sub-conditions, and species. However, I believe the paragraph needs clarification. At first sight, it is hard to find where the 144 trials come from. I would suggest the authors do something similar to what they are already doing in the Statistical Analysis section (e.g., information in brackets: 4 species X 3 conditions X 3 sub-conditions X 4 variations for a total of 144 trials).

R5: We included this information in the first sentence. Also for better flow, we positioned the sentence about controlling for size and luminance further down the paragraph.

Q6: Out of curiosity, I wonder if researchers considered including human faces to control for familiarity given how unfamiliar they were to all animals except conspecifics.

R6: These chimpanzees are not unfamiliar with the animals in question. They are quite familiar with rodents in their enclosures, and outside them to a lesser extent cats and lesser still dogs. The issue with using human faces is that they look fairly similar in arrangement to chimpanzees. Another issue is that the adult human skull resembles an young chimp skull (another reviewer mentioned using infant chimpanzee skulls to control for size). This becomes problematic even if they show more interest in the human stimuli since we cannot disentangle the question of whether they are treating them as chimp-like skulls or recognize the familiar characteristics of human faces in the skulls they see. It was beyond the scope of this study to determine effects such as expertise vs configural processing of face-like stimuli, but definitely something to consider in future studies.

Q7: I found a bit unclear which conditions did they received on each testing day. For example, were face and skull conditions always presented together with stones? In other words, days with face and stone and days with skull and stone? I believe this is not the case, and every day they received either skulls or faces or stones, but I am not entirely sure.

R7: We included a new figure detailing the presentation of the stimuli on each trial and made it more clear it in the text.

Q8: Statistical Analysis

I was puzzled that the authors did not statistically analyze whether chimpanzees look significantly longer to faces vs. skulls vs. skull-shape stones. For instance, if chimpanzees' average looking time is not significantly longer for skulls than skull-shape stones, the significance of the overall results may change. To do that, the authors could try LMM with a Gaussian error structure. A linear mixed model would allow authors to study in the same model the effects of species, condition (skull, face, skull-shaped stone) and sub-conditions together with any interaction between these variables. The authors could also control for any learning and order effects by including session and/or trial numbers. Such a model would also allow authors to not just control for subject ID as a random effect (as in simple ANOVAs) but also the random slopes between the random effect (subject) and the main effects (e.g. condition, species, sub-conditions). The package the authors use is suitable for the proposed analysis.

Notice, though, that I am not stating that the current analyses are incorrect. On the contrary, the authors could extract more information from their data while better controlling the effect of specific variables. Furthermore, the authors would reduce the amount of ANOVAs (eight if I am not mistaken) to just a couple of models focusing on overall looking times and on specific regions of interest within the chimpanzee sub-sample.

R8: We followed this advice and reworked our analyses to include these interactions by type (skull, stone, face).

Q9: Results

Overall looking time

I am not entirely sure why the authors need the first paragraph of the result section. If this is a requisite of the journal, it is ok, but it feels like part of the discussion. It is anticipating the results that are later presenter in greater detail with statistical values.

R9: We removed the paragraph.

Q10: This first section also needs some rephrasing in any case. The authors mention that there appears to be a significant trend across skull face and stone conditions regarding stimuli in frontal sub-conditions. This is right, but then I do not understand why right after this sentence, the authors reiterate that there was more substantial attention to diagonal and frontal (again) sub-conditions in both skull and face but diminished in the lateral sub-conditions. I think it is better to first present the effect on the frontal sub-condition at once.

The overall looking time section seems to introduce the results. I would clarify here why the authors only considered diagonal and frontal chimpanzee faces and skulls for investigating the stimuli chimpanzee look at. The authors only explain that by the end of the results section.

R10: We addressed this by removing the paragraph (see the previous answer). As the reviewer suggests, we've now organized the results by angle and not by type.

Q11: Skulls and faces

In these two results sections, the authors mention, by the end of both paragraphs, that "all other comparisons were non-significant". Do the authors refer to comparisons within the lateral sub-conditions or in general? The statement seems to be especially unclear for the face sections.

R11: We reworked this section completely, we hope it's an improvement.

Q12: Teeth and mouth

I wondered what was the reason not to show animal faces (or at least chimpanzee faces for this analysis) with the mouth open so that teeth were visible? Was it due to the potential emotional reactions they elicit (e.g., fear grin face in chimpanzees)? I understand the authors cannot control for eye presence in skulls, but it is possible to show teeth in alive faces. It might be worth clarifying the reasons for future readers.

R12: To avoid possible matching the teeth through the images themselves (skulls and faces) we only used neutral faces. A second reason was while grinning faces appear in both cats and dogs these don't seem applicable to rats. We included the first explanation in the methods section.

Q13: Discussion

General findings

The sentence starting with "Moreover, compared to..." is very unclear. I would recommend reformulating it. Similarly, the sentence discussing the non-significant difference between chimpanzees' and cats' frontal skulls is hard to follow.

R13: We rewrote this sentence, hopefully, it is clearer now.

Q14: The authors state that for the lateral skull sub-condition, there were no significant differences among stimuli. Please correct the statement. They seem to look longer for rats than all other species—and there are significant differences between rats and dogs.

R14: This was likely connected to peculiarities of the rat skull in the lateral angle (mouth looks open as if vocalizing or biting). In the newer analyses, these effects are no longer visible.

Q15: The end of the second discussion paragraph needs references.

R15: We added the additional sentences at the end: *“Our findings are in line with research in face pareidolia (detection of illusory faces on non-living objects) in humans (Liu et al., 2014), chimpanzees (Tomonaga, 2013), and rhesus monkeys (Taubert et al, 2017) where they perceive faces in inanimate artifacts. The chimpanzee skull, we argue, falls within these highly pareidolic objects.”*

Q16: Further considerations

The end of this section is slightly confusing. It needs rephrasing (e.g., ..they previously seen dead..).

R16: We changed it to “There remains the issue of whether chimpanzees in the wild associate skeletons together with places where dead individuals were previously seen.”

Q17: Conclusions

The sentence starting "For H1;.." needs rephrasing.

R17: We removed the extraneous wording.

Q18: For H2, why is the non-significant trend between chimpanzees and rats' frontal faces likely to disappear with a larger sample size? Curious to know why this is the only trend interpreted that way.

R18: It is our opinion that some of the interest was due to individual differences. Should we increase the chimpanzee subjects possibly this effect would disappear. It was a conjecture with no strong support so we removed it.

Q19: Finally, by the end of the manuscript (page 9) the authors discuss that the hypothetical face-module develop within the context of frontal face-to-face interactions for the first time. Before, they only refer to this module in general terms. Could you elaborate why it evolved in the context of face-to-face interactions? As of now it reads as a conclusion stemming solely from your results.

R19: We elaborated on this point in section 5.2, there a passage reads: *“According to Senju and Hasegawa (2005) direct gaze signals intention towards the receiver (i.e. communicative, affective, hostile, friendly, or sexual) making it adaptative to direct attention towards frontal faces. This ability follows a developmental trajectory that starts with frontal faces being processed in dedicated areas of the brain, and somewhere around 8 months of life profile faces are integrated into the same brain regions (Nakato et al., 2009).”*

Reviewer: 5

Comments to the Author(s)

The authors present a novel eye-tracking study in chimpanzees that examines visual attention to pictures of faces, skulls, and stones (that mimicked the shape/color of the species). Four different species of mammals were presented: chimpanzee, cat, dog, and rat, and each condition was presented at three different angles (diagonal, frontal and lateral). As such, this is an impressive dataset for examining heterospecific and conspecific face perception in chimpanzees along three main lines of inquiry:

- 1) looking times to conspecific versus heterospecific faces (which is not in itself novel, but the inclusion of these particular mammalian species is),
- 2) whether looking times change with angle of presentation and if so, whether the changes are consistent across con/heterospecifics or not,
- 3) whether progressive 'degrading' of the face signal (from face to skull to stone) alters looking times, and whether this is consistent across con/heterospecifics,
- 4) what are the salient features of the presented cues.

Q1: Unfortunately, rather than using the theoretical framework of face processing here, the authors attempted to ground their research in comparative thanatology, which is not the most appropriate framing here. Consequently, I do not think this manuscript is publishable in its current form. If the authors were willing to substantially revise the manuscript, including the theoretical framework, then I think it could be a publishable contribution. The manuscript should also be edited for clarity/word choices throughout. There are some unnecessarily long and confusing sentences (e.g. the last sentence of 1. Summary).

R1: We have substantially reworked the discussion to include relevant comparative facial research literature, we firmly believe this made the manuscript more firmly grounded in this field while still acknowledging the former field. Throughout the text we strove to simplify overlong sentences to facilitate the reading and its general flow.

Specific comments:

Q1: Introduction:

As described above, the thanatological framework is weak. The authors themselves note that it is extremely rare for chimpanzees to come across conspecific skulls, so I am a bit confused as to why they chose to justify this study in this manner. While this is worth some treatment in the discussion, there is a far more rich literature on how chimpanzees and other primates process faces that provide appropriate background and justification for the study including:

For comparing looking times to con- and heterospecific faces: 1) Kano and Tomonaga 2009 - chimpanzee viewing patterns when shown pictures of chimpanzees, humans, other mammals; 2) Hattori et al. 2010 – differential sensitivity to conspecific and allospecific cues in chimpanzee and humans

Angles of presentation/degradation of signal: 1) Tomonaga & Imura 2009; 2) Dahl et al. 2013; 3) Taubert et al. 2012; 4) Herman et al. 1990 (in dolphins!)

Face scanning/salient features – many, but including: 1) Hirata et al. 2010; 2) Kano et al. 2015; 3) Kano et al. 2012

R1: We included much of suggested comparative facial literature and more and reworked section 5.2.

Q2: Methods:

Please provide more description on calibration methods, I do not know what “two small clips on each referent point” means.

R2: We changed the sentence to “these involved one small clip of a stirring object presented twice on each opposing corner of the screen”.

Q3: Figure 1. is quite confusing since the stimuli in front of the chimpanzee in the figure looks nothing like the examples provided in Figure 2, please clarify.

R3: We added a figure charting the stimuli presentation of each trial. Note that we used the same diagonal skull images as featured in Fig1 in this new figure. Note also that there were four sets of skulls per species and per orientation (the diagonal skulls presented in Fig1 and Fig2 are two out of four skull variations). We explained this more clearly in the methods.

Q4: Figure 2. Please rework this to show the species in the same configuration across all conditions (e.g. chimpanzee top left, dog top right, cat bottom left, rat bottom right). I understand that this is not how they were presented to subjects (to avoid positional biases), but it would be useful to be able to match the stone condition directly to the species/angle of interest.

R4: We changed the order accordingly.

Q5: More information needs to be provided on the statistical methods. What does AOI (species-stimuli) refer to? Is this, for example, cat-face (vs cat-skull) or cat-frontal versus cat-lateral? It seems like it is the skull/face/stone category differences later in the paragraph, but then I don't understand how the angle of presentation was analyzed. Please clarify this so that your work is replicable.

R5: We rewrote the methods and results section. These points are now made clearer in the manuscript. We ran three separate analyses for each angle to see the interactions between species and type.

Q6: Results:

Please provide descriptive statistics (mean looking times and sds) for all comparisons.

Please also refer to the appropriate figures in the text of the results.

A summary table of results would be much more digestible.

R6: We included these tables in the manuscript now.

Q7: Discussion:

The discussion should be reworked according to the suggested framing above.

R7: We have reworked the discussion accordingly, thank you for the suggestions.

===PREPARING YOUR MANUSCRIPT===

- one version identifying all the changes that have been made (for instance, in coloured highlight, in bold text, or tracked changes);
- a 'clean' version of the new manuscript that incorporates the changes made, but does not highlight them. This version will be used for typesetting if your manuscript is accepted.

===PREPARING YOUR REVISION IN SCHOLARONE===

Please ensure that you include a summary of your paper at Step 2 'Type, Title, & Abstract'. This should be no more than 100 words to explain to a non-scientific audience the key findings of your research. This will

be included in a weekly highlights email circulated by the Royal Society press office to national UK, international, and scientific news outlets to promote your work.

-- Ensure that your data access statement meets the requirements at <https://royalsociety.org/journals/authors/author-guidelines/#data>. You should ensure that you cite the dataset in your reference list. If you have deposited data etc in the Dryad repository, please include both the 'For publication' link and 'For review' link at this stage.

Appendix B

Reviewer: 4

The authors have significantly improved an already interesting manuscript during the review process. However, I still have a few comments about the interpretation of the results that would need clarification before my final acceptance.

Q1: My main comment concerns the interpretation and analysis of the results. It is important to show that chimpanzees fixated significantly more on chimpanzee faces than on chimpanzee skulls. This would support the authors' interpretation stating that chimpanzees may interpret skulls as degraded faces. As of now, it is not easy to see that this has occurred. The interaction effect between type and species in the frontal and diagonal models supports that chimpanzees fixated more in conspecifics than in other species (with some exceptions) for frontal and diagonal faces and skulls, but not for stone-shaped images. But there seems to be no apparent difference in their fixation patterns for chimpanzee faces vs. skulls with other species. One possibility to check faces > skulls > stones (at least in chimpanzees) is to include chimpanzee species and test the main effect of type. Then, using the lme4 package, the authors could use the drop1 function to assess pair-wise comparisons between type levels. Re-leveling the model variables (e.g., species) and using the drop1 function might be better to compare chimps against other species instead of using the Bonferroni correction.

R1: We've now addressed this issue. The main idea that it was important to show that chimpanzees fixated significantly more on chimpanzee faces than chimpanzee skulls or stones was tested in a separate experiment. All the experiments can now be seen in Figure 4. We did it since it best followed the reviewer's intent but also comparing the stimuli with data from the previous experiment was probably not the best practice and, more critically, it allowed for a direct comparison between these three chimpanzee types. We've used the drop1 function throughout along with the Bonferroni corrections for the levels in each variable since drop1 only compared variables. That being said, for additional transparency, we've included all the model summary results from all the analyses in the supplementary data with species "chimp" and type "face" re-leveled (in Tables 2, 3 and 4).

Q2: Page 19 of 55 (the document I got from RSOS), L 16: In section 3.3, it is unclear that apes experienced only one type of stimuli (face, skull, or stone) within a single image group. The authors wrote, "each image group had three types.", however, it seems that each image group was composed of four pictures, one per species, and all of them were of the same type (either face, skull, or stone)—as shown in Figure 2.

R2: Thank you for pointing out this oversight, we changed the sentence accordingly it now reads: *"Each image group consisted of one of three types (either skull, face, and or skull-shaped stone)"*

Q3: Page 19 L 48: When you refer to three independent tests (one for each angle), do you mean three independent LMM models, correct?

R3: Correct, we changed the sentence accordingly.

Q4: Page 20 L 7: The presentation of the results is slightly inconsistent across sections. For instance, when the authors state that the "fixation durations were significantly longer overall for the chimpanzees," I would instead state that the fixation durations were significantly longer for

chimpanzees compared to or versus other species. The authors already do something similar to what I propose in L 30.

R4: We've reworked these three sections to read more consistently across each other. Thank you.

Q5: Page 21 L 42: Either use *versus* or versus.

R5: Thank you for noticing this, all versus words are now italicized.

Q6: Page 21 L 48: The sentence needs to be rephrased "this suggests the stones, with only outlines of each species, were too degraded a stimuli..."

R6: We changed the sentence, now it reads: "This suggests the stones, showing only the outlines of each species, were a too degraded facial stimulus-type to retain their interest"

Q7: Page 21 L 55: The authors suggest that fixation times were slightly higher in diagonal faces compared to diagonal skulls. Do they refer to fixation times for all stimuli or just for chimpanzees? This is the kind of comparison that would support the faces > skulls > stones, at least for diagonal angles, and it would be interesting to know whether those differences are significant. Table 3 seems to support this for species chimpanzee, although there is a high variation for chimpanzee diagonal faces fixation times.

R7: We've now conducted an additional experiment that directly compares chimpanzee stimuli with the different types (face, skull, stone) together and included those results right after the lateral results.

Q8: Page 22 L 8: I would suggest the authors include the unpublished results in the Electronic Supplementary Material. This analysis supports the authors' interpretation of the lateral fixation times for rat skulls.

R8: We have now included these results in the supplementary material, along with graphs and an additional figure showing the uncovered and the covered rat skull.

Q9: Page 23 L 10: The sentence needs some revision "one possible explanation is the result was due..."

R9: We revised the sentence now reading: "*One likely explanation for the discrepancy is the differing methodologies employed by each study (i.e. 1 large face per trial versus 4 smaller faces per trial)*"

Q10: Page 25 L 19: The sentence in brackets is not completely accurate. The frontal conditions had overall longer looking times than diagonal conditions, but for instance the fixation time for diagonal chimpanzee face is longer than for frontal chimpanzee face.

R10: To be fair, we used the \approx symbol (similar) rather than the = symbol (equal). We also moved the brackets to the introduction (expected results) paragraph of that section, instead of the actual results paragraph below. This section has also been rewritten.

Reviewer: 5

Comments to the Author(s)

This revision of the submitted paper is much improved in terms of clarity, especially with regards to the statistical approach and the results. However, I still think it is too heavily framed in the introduction as a thanatological study (which it is not) versus a face perception study.

Q1: Aside from the abstract/summary, a reader of the introduction as written would assume that they were about to read about a study in which various skulls were placed in a testing area of the PRI and chimpanzees attention to and interaction with those skulls was measured. It is not until the second paragraph of section 2.3 that the reader becomes aware that this is an eye-tracking study, at which point, many readers would think, “Wait, there is a large literature on conspecific and cross-species face perception, including using eye-tracking, so how is this study situated in that literature?”. My recommendation would be to completely reframe the introduction to focus on the face-processing and eye-tracking literature that is now not reviewed until the discussion. The link to thanatology is intriguing, but this study does not tell us anything about what chimpanzees know, feel or do in response to death. Given that the main results are 1) chimpanzees prefer to look at chimpanzee faces over dog, cat and rat and 2) that this preference transfers to a partially degraded face signal (skulls), but not to a more degraded signal (stones), and 3) teeth seem particularly salient, I think this paper is more appropriate for a field specific journal, such as *Journal of Comparative Psychology* or *Animal Cognition*.

R1: The reviewer is correct: indeed this is a face perception study using similar methodology to other face perception studies, and with a close connection to face pareidolia research, this we do not dispute. But we would argue, for its implications, it is also a comparative thanatology study, since we are using skulls. Our interest was to tie these two lines of research together, we could have used any other face-like stimuli (chimpanzee-face shaped toasts), and the results might have been similar. The point of using chimpanzee skulls is that they are death-related stimuli which are found in nature. Presenting skulls to chimpanzees (*sensu* McComb et al., 2006), or observing chimpanzees in the wild reacting to skulls and skeletons (see Watts, 2020) would always have a limited explanation as to why they are showing this interest and certainly little about how they “feel” about it. With this study, we can say there is a strong indication for facial component to the skull guiding their attention, which one can argue with other cues might anchor or serve as a basis for chimpanzees’ knowledge about death. Without using skull stimuli there would be no point (to us) in this study being tied with comparative thanatology. We still hope it is an interesting/adequate study enough to publish in this Journal, since the reviewer and the other four reviewers have kindly given their time and energy making critical insights that contributed in shaping the manuscript into its final form.

Section and line specific comments are below:

Q1: 1. Summary (Abstract) This reads as if it is prefacing two studies – one about thanatology and one about face processing

R1: We have now removed the elephant study from the abstract to focus only on our experiment.

Q2: Lines 32-33 – Use “hypothesize” or “predict” rather than “suspect” and then state your predictions clearly, otherwise the sentence that begins “supporting our hypotheses” is unclear.

R2: Thank you for pointing this out, we removed “suspect” and included “hypothesized” instead.

Q3: Line 33 – I would use “preferred” or “attended to” rather than “attracted”

R3: We changed it to “preferred”.

Q4: Line 34-35 – In the sentences with numbered results add the direction of the effect “chimpanzees attended significantly MORE towards conspecific skulls”

R4: We added the direction of the effects in all numbered results.

Q5: Line 57-59 – I would remove the last sentence of this paragraph and just add the Halder reference to line 54 as so (Hutson et al., 2013, but see Halder & Schenkel, 1972).

R5: There are not many studies like McComb et al. 2006 which is the major inspiration for our paper. An exception is Halder & Schenkel, an obscure study written in German particularly aimed at bovid’s responses toward skeletons. Since a previous reviewer asked for us to specify the species studied, we decided to keep it. We also find it helpful to uncover the main findings from this paper as it relates to McComb’s and ours.

Q6: Introduction, Section 2.2 (page 3) Line 31 – Insight into what question?

R6: There was a sentence before this one we removed at the request of a reviewer. We realize now we could have made the following sentence clearer. We changed it to: *“Two classic comparative psychology studies give us some insights into this topic regarding skeletons.”*

Q7: Line 32 – “researched the comparative development of chimpanzees and humans” - you do not need to state an author’s field of study.

R7: We only included that part to situate her experiment with dead animals she did with her chimp. We also state the same for Donald Hebb; his experiments were on the phylogeny of fear.

Q8: Line 44-46 – This sentence leaves out an important category of chimpanzee reactions, namely maternal responses to corpses (Biro et al. 2010, Lonsdorf et al. 2020, Carter et al. 2021)

R8: We included an extra sentence to account for dead infant carrying, it now reads: *“Indeed, chimpanzee mothers have been observed carrying their dead infants for days, weeks or months (Lonsdorf et al., 2020), a pattern commonly observed in many females across the primate order (Watson & Matsuzawa, 2018; Fernandez-Fueyo et al., 2021).”* Carter’s paper deals specifically with chacma baboons, but we included Fernandez-Fueyo, Sugiyama, Matsui & Carter 2021. Biro’s paper appears in later sections of our manuscript.

Section 3.3 Procedure

Q9: Page 5, Line 20 - I know that face-processing researchers are concerned with controlling for the size of stimuli. However, in this case, you should consider prioritizing ecological validity given that you are trying to gain insight into a process that happens in nature. That is, chimpanzee faces, and skulls are naturally bigger than cat, mouse and (depending on the breed) dogs, so it’s possible that the chimpanzees were attentive to chimpanzees faces and skulls due to the novelty of those items

being presented much smaller than the natural size, or at least the natural proportional size to a mouse face/skull.

R9: The reviewer makes an interesting point. From the screen at the resolution these were presented, the chimpanzee skull images (in experiment1) would have had a size of a between a rhesus monkey's skull and an adult sized chimpanzee skull. The dog would also have a reduced size compared to the actual size (i.e. German Shepard, Siberian husky) while the cat and the rat skulls were enlarged compared to their actual skulls. The same applies to face images. This *size-novelty argument* can be made for species faces with which the chimpanzees in our study are familiar (to some extent). But considering *size-novelty* a bit deeper, are not dog faces (being smaller) and cat and rat faces (being larger) just as novel? And why do they show the same looking trend for chimpanzee skulls with which they have never come in contact, thus having no frame of reference? We assume the reviewer meant that aside from a bias toward species, there could also be a size bias simultaneously at play and not one in detriment to the other. We suppose one way to test if size affects the chimpanzees' attention regardless of species or type, could be to manipulate the images for which in one case the chimpanzee (face/skull) is the larger stimuli and in other cases the rat is the larger and so on. One of our arguments was that McComb's study with the elephants would have been stronger had they used younger/infant elephants' skulls of equal size to other species' skulls. We do think elephants recognize to some extent their species skulls, but we could see a leaner counterargument being made with regards to the larger size as the main cue. We don't dispute the reviewer's ecological validity point, so we added this thought to our future directions section of the conclusion.

Q10: Results/figures. In figures 2, 4, 5, 6, I recommend reordering the X-axis so that face, skull, stone, is the order going from left to right, which follows the path of degradation

R10: Great suggestion, it does make a more elegant presentation of the data thank you, we also did it for the stimuli figure.

Q11: Discussion, Section 5 Page 7 Line 19 – Missing word between “find” and “if”.

R11: We changed it to “*was to find out if chimpanzees*”

Q12: Line 26-27 – If I am understanding your statistical results correctly, then you cannot say that there was any significant preference for conspecific skull-shaped stones.

R12: Good point, we clarified this in the sentence: “*Moreover, this was particularly evident in frontal/diagonal orientations for chimpanzee faces and skulls.*”

Q13: Page 8 Line 19-22 – Yes, this is the main finding, that chimpanzee skulls are likely detected as degraded faces. I recommend rewording this as skulls do not present themselves.

R13: We removed themselves from the sentence now reading: “*chimpanzee skulls, essentially, exhibit a degraded signal of a face*”.

Q14: Section 5.2 – much of this belongs in the Introduction, especially the first four paragraphs. Line 36-50 – This paragraph on neuroimaging findings seems outside the scope of this paper, I recommend excluding it.

R14: We included half of this section in the Introduction now. We respectfully disagree on the neuroimaging findings, especially since some of these relate to face pareidolia research tied to this

paper and further down in the Conclusion section. Also, our manuscript has “face-module” in the title, this claim would become substantially weaker without support from neuroimaging studies.

Q15: Section 5.3 – while interesting, does not have much relevance for what was actually tested. The only portion that is relevant for this paper is the paragraph on tusks and teeth, but this could be much reduced and included in the face-processing portion.

R15: We thought it important to delve into the ecological validity of our study, tying into the implications of our findings to wild chimpanzee populations. This is related to our previous claim that this is not just a face research paper for face perception researchers but also directed at field researchers from which we took inspiration.

Q16: Conclusion & Future Directions, Page 11 I found these two paragraphs much more relevant and explicitly linked to the experimental paradigm, and therefore much more compelling.

R16: We sincerely thank the reviewer for the valuable comments!